# A combined experimental and modelling approach for the Weimberg pathway optimisation

Lu Shen[1], Martha Kohlhaas[2], Junichi Enoki[3], Roland Meier[4], Bernhard Schönenberger[4], Roland Wohlgemuth [4,5], Robert Kourist [3,6], Felix Niemeyer [2], David van Niekerk[7], Christopher Bräsen[1], Jochen Niemeyer [2✉], Jacky Snoep [7,8✉] & Bettina Siebers [1✉]

The oxidative Weimberg pathway for the five-step pentose degradation to α-ketoglutarate is a key route for sustainable bioconversion of lignocellulosic biomass to added-value products and biofuels. The oxidative pathway from *Caulobacter crescentus* has been employed in in-vivo metabolic engineering with intact cells and in in-vitro enzyme cascades. The performance of such engineering approaches is often hampered by systems complexity, caused by non-linear kinetics and allosteric regulatory mechanisms. Here we report an iterative approach to construct and validate a quantitative model for the Weimberg pathway. Two sensitive points in pathway performance have been identified as follows: (1) product inhibition of the dehydrogenases (particularly in the absence of an efficient $NAD^+$ recycling mechanism) and (2) balancing the activities of the dehydratases. The resulting model is utilized to design enzyme cascades for optimized conversion and to analyse pathway performance in *C. cresensus* cell-free extracts.

[1] Molecular Enzyme Technology and Biochemistry (MEB), Environmental Microbiology and Biotechnology (EMB), Centre for Water and Environmental Research (CWE), University of Duisburg-Essen, Universitaetsstrasse 5, 45141 Essen, Germany. [2] Institute of Organic Chemistry and Center for Nanointegration Duisburg-Essen (CENIDE), University of Duisburg-Essen, Universitaetsstrasse 7, 45117 Essen, Germany. [3] Junior Research Group for Microbial Biotechnology, Ruhr-University Bochum, Universitaetsstrasse 150, 44780 Bochum, Germany. [4] Member of Merck Group, Sigma-Aldrich, Industriestrasse 25, Buchs 9471, Switzerland. [5] Institute of Molecular and Industrial Biotechnology, Technical University Lodz, Stefanowskiego Street 4/10, Lodz 90-924, Poland. [6] Institute of Molecular Biotechnology, Graz University of Technology, Petersgasse 14, Graz 8010, Austria. [7] Department of Biochemistry, University of Stellenbosch, Private Bag X1, Matieland 7602, South Africa. [8] Department of Molecular Cell Physiology, VU University Amsterdam, De Boelelaan 1085, 1081 HV Amsterdam, The Netherlands. ✉email: jochen.niemeyer@uni-due.de; jls@sun.ac.za; bettina.siebers@uni-due.de

Lignocellulosic biomass is considered a promising renewable resource for 'second generation' biofuel and added-value chemical production without affecting food supply[1]. The complex and recalcitrant structure poses challenges in its conversion[2,3] and pretreatment of (ligno)cellulose by physico-chemical and enzymatic methods is necessary, after which the main fermentable constituents of the hydrolysate are D-glucose and D-xylose. Whereas common biotechnology platform organisms (such as *Escherichia coli* and yeast) can convert D-glucose efficiently into valuable products, their metabolic capabilities for efficient bioconversion of pentoses and for co-fermentation of hexose/pentose mixtures are rather limited. Therefore, many metabolic engineering and synthetic biology approaches aim for efficient conversion of D-xylose to bioproducts using whole-cell biocatalysis[4,5] (for reviews, see refs. [6,7]).

In that respect, the oxidative Weimberg pathway (Fig. 1a) has gained major attention[8]. The pathway has been identified in several bacteria and archaea[9–12], and is best understood for the oligotrophic freshwater bacterium *Caulobacter crescentus*, in which the involved enzymes are encoded in the D-xylose-inducible *xylXABCD* operon (CC0823-CC0819)[13] (Fig. 1b). In this pathway, D-xylose is first oxidized by D-xylose dehydrogenase (XDH) to D-xylonolactone, which is then hydrolysed to D-xylonate either

non-enzymatically or by the xylonolactonase (XLA). In two steps of dehydration, the D-xylonate is converted via 2-keto-3-deoxy-D-xylonate (KDX) to α-ketoglutarate semialdehyde (KGSA) catalysed by D-xylonate dehydratase (XAD) and KDX dehydratase (KDXD), respectively. In the last step, KGSA is oxidized in an NAD(P)$^+$-dependent manner by KGSA dehydrogenase (KGSADH) to finally yield α-ketoglutarate, a key intermediate in the citric acid cycle[8].

For the biotechnological conversion of lignocellulose-derived pentoses, the Weimberg pathway offers several advantages compared with the other pentose degradation routes, as: (i) no ATP is required for sugar activation, (ii) the oxidative pathway is thermodynamically favourable, i.e., all reactions have highly negative standard Gibbs free energy changes ($\Delta G^{0'}$ values) (Fig. 1c)[14], (iii) no carbon loss occurs at the level of pyruvate and isocitrate conversion to acetyl-CoA and α-ketoglutarate, respectively, (iv) the pathway is rather short compared with other pathways and the intermediates do not branch off into other central metabolic routes, and (v) the product α-ketoglutarate is a good starting point for biosynthesis of valuable compounds.

Therefore, the Weimberg pathway (or parts thereof) from *C. crescentus* have already been employed in several metabolic engineering and synthetic biology approaches, e.g., for production

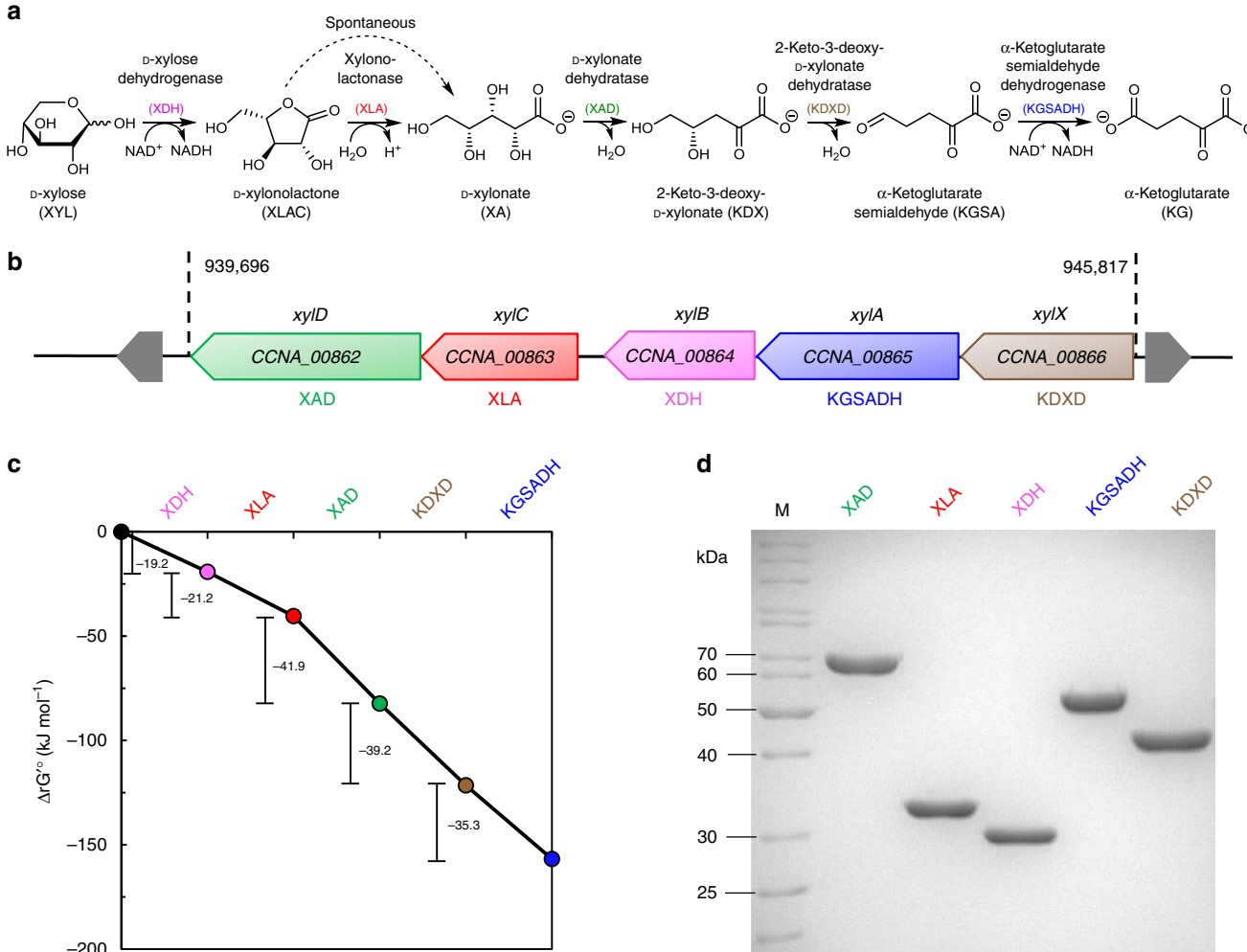

**Fig. 1 The Weimberg pathway for D-xylose conversion in *C. crescentus*.** The individual reactions of the pathway and the corresponding enzymes catalysing them (**a**), the D-xylose operon with the respective genes (*xylBCDXA*, *C. crescentus* NA1000) and encoded proteins (**b**), the change in standard Gibbs' free energy for the oxidative Weimberg pathway[14] (**c**), as well as the recombinant proteins after purification (SDS-PAGE and Coomassie staining) (**d**) are shown. For D-xylonolactone, the spontaneous, non-enzymatic conversion to D-xylonate is indicated (dotted line) (**a**). Abbreviations for enzymes and intermediates are given in the Fig. 1a; M, molecular weight marker. The source data underlying Fig. 1c are provided as a Source Data file.

of 3,4-dihydroxybutyric acid, 1,2,4-butanetriol, 1,4-butanediol, ethylene glycol and glycolic acid (for a recent review, see ref. [7]). For whole-cell biocatalytic conversion of D-xylose to D-xylonate, the XDH and XLA were introduced into *Corynebacterium glutamicum*[15], *E. coli*[16,17] and *Saccharomyces cerevisiae*[18]. The complete pathway for oxidative D-xylose degradation was introduced into *E. coli*[19], *Pseudomonas putida*[20], *C. glutamicum*[21–24] and *S. cerevisiae*[25,26]. However, the results obtained so far in these whole-cell bioengineering approaches have been suboptimal and indicate that intermediate accumulation (i.e., accumulation of toxic D-xylonate) and poor protein expression (e.g., XAD) might be inhibitory to host metabolism and product formation. Therefore, adaptive laboratory evolution (LAE) approaches[22,26], introduction of additional gene copies[26,27] and approaches in which the number of genes have been reduced (possible because previously unknown endogenous enzyme activities catalyse the reactions)[20,23] have been followed.

To circumvent potential problems caused by cellular complexity[28], in-vitro enzyme cascades have been used as an alternative to whole-cell biocatalysis. Compared with chemical approaches, enzyme cascades enable the production under moderate, environmentally friendly conditions and, most importantly, provide stereo-selectivity for product formation[29,30]. However, although much simpler than intact cells, one-pot enzyme cascades function similar to metabolic pathways and are subject to regulatory properties (e.g., inhibition by pathway intermediates, (co-)substrates and products) and need quantitative tools for analysis. To identify the rate-limiting step(s) and to optimize enzyme cascades, currently mainly combinatorial approaches are used with non-biased combinations of cascade elements and/or reaction conditions (e.g., buffer composition, pH and temperature)[31–33] (for reviews, see ref. [30]). Rational approaches involving computation and modelling have so far rarely been included[34–37] and the results for computational pathway design are often hampered by missing data for enzyme properties and incomplete information of systems behaviour. Thus, in most cases, the system cascade properties are not fully understood and bottleneck identification remains difficult, which hampers pathway optimization and design.

For the Weimberg pathway in *C. crescentus*, enzyme kinetic data are available for three of the five enzymes: XDH, XAD and KDXD, and for XAD also the crystal structure was solved[38–42]. However, product inhibition or inhibition by Weimberg intermediates has not been addressed so far and, to our knowledge, no kinetic model for the pathway has been reported.

Therefore, in this study, we develop an iterative multi-step approach: initial rate kinetics on isolated enzymes and in-vitro enzyme cascades for model construction, and cell-free extract conversion assays for model validation and application, yielding a quantitative kinetic model that precisely simulates the Weimberg pathway under various conditions. Using computational design, the in-vitro enzyme cascade is optimized for the highest conversion efficiency and the in-vivo pathway performance in

*C. crescentus* cell-free extracts is analysed. Thus, we can reproduce many of the problems encountered in metabolic engineering projects of the Weimberg pathway and importantly indicate possible solutions for these problems, demonstrating the broad applicability of the Weimberg model. Finally, we show two biotechnological applications of the enzyme cascade to illustrate the industrial relevance.

## Results

**Weimberg enzyme cascade and model construction.** The first and most basic step in the development of the cascade model was to precisely characterize the single enzymes under physiological conditions to provide initial rate kinetics. Therefore, the five genes of the D-xylose operon of *C. crescentus* were cloned, expressed in *E. coli* and the recombinant enzymes were purified (Fig. 1d). The Weimberg enzymes were characterized in detail in terms of kinetic properties, substrate specificity, effect of buffer composition, pH and temperature optima, as well as suitable storage conditions (Supplementary Tables 1 and 2, and Supplementary Figs. 1–3). The metal ion dependency of the XAD was confirmed and further investigated. Among different divalent metal ions, $Mn^{2+}$ was the most effective in restoring enzyme activity (Supplementary Fig. 2). Therefore, $Mn^{2+}$ was added at low concentrations (17 μmol $MnCl_2$ per mg protein) to the XAD protein solution and in further assays it was ensured that the presence of metal ions did not interfere with the other enzyme activities and later nuclear magnetic resonance (NMR) analysis. The final determination of kinetic parameters was performed in HEPES buffer at pH 7.5 at 37 °C. For all five enzymes, a Michaelis–Menten-type rate equation, based on a random-order binding mechanism with irreversible kinetics, was fitted to the experimental data to yield maximal rates ($V_M$) and binding constants for the substrates and cofactors (Table 1, Supplementary Note 1, Supplementary Tables 3–7 and Supplementary Figs. 4–8).

Next, we performed a progress curve analysis in the sequential enzyme cascade. Therefore, based on the initial rate kinetics, a model was developed for the sequential conversion of D-xylose to α-ketoglutarate and was used for experimental design (choosing appropriate enzyme concentrations), to ensure that each reaction converts 5 mM substrate completely to product in 90 min, which is suitable for NMR analysis. The NMR analysis ($^1H$-NMR and $^{13}C$-NMR, enabled by the use of D-xylose-1-$^{13}C$) allowed for a time-resolved observation (1 data point in $^1H$ and $^{13}C$ every 5 min) of the respective pathway intermediate concentrations (Supplementary Note 2, Supplementary Data 1 and Supplementary Figs. 13–27). All of the enzymatically produced intermediates were in agreement with the proposed Weimberg pathway[13]. However, for several intermediates (i.e., D-xylose, KDX, KGSA and α-ketoglutarate), isomers and/or hydrates were observed (Fig. 2a, b). In accordance with the thermodynamics (eQuilibrator database[14], http://equilibrator.weizmann.ac.il/; Fig. 1c), all

**Table 1 Initial rate kinetics of the D-xylose degrading enzymes from *C. crescentus*.**

| Enzyme | Substrate/cofactor | $K_m$ (mM) | $V_{max}$ (U mg$^{-1}$ protein) | $k_{cat}$ (s$^{-1}$) | $k_{cat}/K_m$ (s$^{-1}$ mM$^{-1}$) |
|---|---|---|---|---|---|
| D-xylose dehydrogenase (XDH) | D-xylose (XYL) | 0.20 | 120 | 57 | 285 |
|  | NAD$^+$ | 0.16 |  |  | 356 |
| Xylonolactonase (XLA) | D-xylono-1,4-lactone (XLAC) | 0.45 | 944 | 497 | 1104 |
| D-xylonate dehydratase (XAD) | D-xylonate (XA) | 0.79 | 42 | 45 | 57 |
| 2-Keto-3-deoxy-D-xylonate dehydratase (KDXD) | 2-keto-3-deoxy-D-xylonate (KDX) | 0.21 | 107 | 74 | 352 |
| α-Ketoglutarate semialdehyde dehydrogenase (KGSADH) | α-ketoglutarate semialdehyde (KGSA) | 0.02 | 49 | 41 | 2050 |
|  | NAD$^+$ | 0.60 |  |  | 68 |

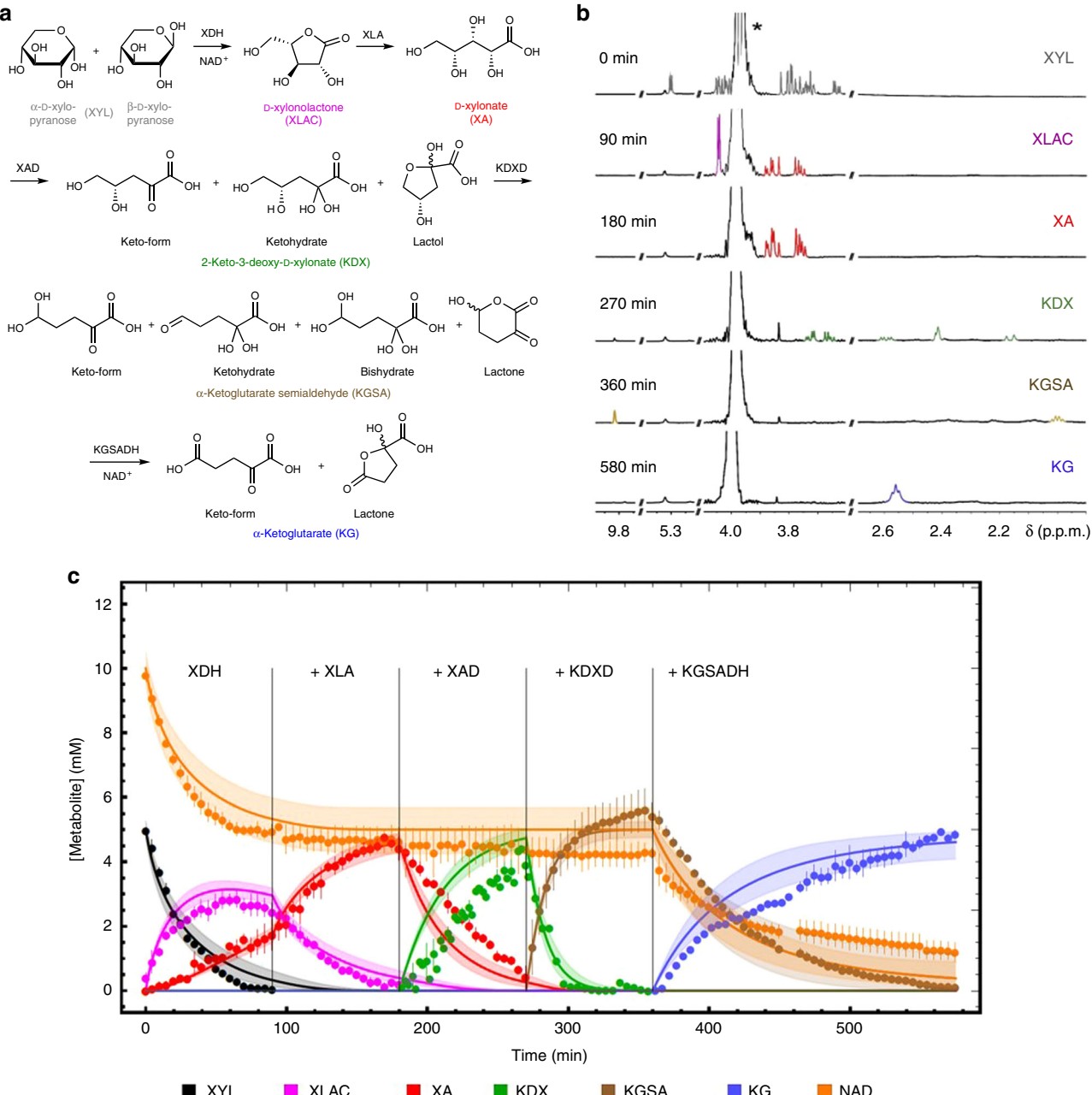

**Fig. 2 Combined progress curves with single enzyme additions.** Conversion of D-xylose to α-ketoglutarate during the sequential addition of the Weimberg pathway enzymes (XDH, XLA, XAD, KDXD and KGSADH added, respectively, at $t = 0$, 90, 180, 270 and 360 min). The different intermediates, i.e., isomers and/or hydrates, determined by NMR (**a**), selected NMR spectra before addition of the next enzyme (*buffer signal) (**b**) and the metabolites followed over time with NMR and the model simulations (**c**) are shown. Two independent experiments were performed (for detailed NMR information, see Supplementary Note 2). A mathematical model based on equations (1) to (6) (Supplementary Note 1) was used to describe the experiment and model simulations are shown in solid lines with corresponding colours to the experimental symbols. Error bars indicate the standard error of the mean (SEM) for the two independent experiments ($n = 2$) and the shaded bands show the solution space for the model simulations with a 10% error margin for the parameters. For abbreviations, see Fig. 1. The source data underlying Fig. 2c are provided as a Source Data file. It can also be accessed via https://doi.org/10.15490/FAIRDOMHUB.1.ASSAY.986.2 or https://jjj.bio.vu.nl/models/experiments/shen2020_fig2c/simulate.

five enzymes catalyse highly exergonic reactions and the reactions ran to completion and resulted in a pure product. The XDH reaction resulted in a mixture of D-xylonolactone and D-xylonate, due to the spontaneous non-catalysed hydrolysis of xylonolactone to D-xylonate.

All reactions completed the substrate conversion in roughly 90 min, which is in agreement with model simulations that include product inhibition (Fig. 2c). However, a much slower

conversion was observed than predicted, based on initial rate kinetics excluding product inhibition (Supplementary Note 1 and Supplementary Figs. 9–11). These results indicated strong product inhibition and, therefore, we extended the initial rate experiments by including products to the assays, wherever possible. Product inhibition constants could be determined experimentally for both dehydrogenases (i.e., XDH and KGSADH) and the first dehydratase (XAD). For XLA and

KDXD, the inhibition constants for D-xylonate and KGSA, respectively, had to be fitted based on the conversion data, as these enzymes are analysed in coupled enzyme assays. The progress curves after single enzyme additions and the required model adjustments are shown and discussed in Supplementary Note 1, all enzyme kinetic parameters are shown in Supplementary Tables 3–7. In Fig. 2c, the carbon metabolites, the cofactor NAD$^+$, as well as the model simulation with product inhibition included are displayed, showing a good description of the data.

As a final step in model construction, the one-pot cascade model was established. The results of the sequential pathway reconstruction were used to define the 'reference state' for the one-pot cascade conversion of D-xylose to α-ketoglutarate (Supplementary Note 1). For this, we added the five enzymes together, using amounts close to those used in the sequential experiment (2.5 μg ml$^{-1}$ XDH, 0.8 μg ml$^{-1}$ XLA, 8.67 μg ml$^{-1}$ XAD, 0.5 μg ml$^{-1}$ KDXD and 10 μg ml$^{-1}$ KGSADH). The conversion of 5 mM D-xylose via the pathway intermediates to α-ketoglutarate was followed using NMR (Fig. 3a). The model based on the isolated reaction kinetics predicted the first part of the pathway accurately, i.e., D-xylose oxidation, lactone cleavage and D-xylonate dehydration. However, conversion in the complete pathway did not run to completion experimentally and, specifically, the conversion via KDXD was slow (high concentrations of KDX remaining). Also, the high KGSA to α-ketoglutarate ratio suggested stronger inhibition of KGSADH than observed previously, even stronger than observed in the sequential experiment. As in this approach all pathway intermediates are present in the reaction mixture at the same time, we tested the sensitivity of the KDXD to all pathway intermediates in the initial rate kinetic experiments (Supplementary Fig. 7 and Supplementary Table 6). These analyses revealed that especially D-xylonate, KGSA and α-ketoglutarate had an inhibitory effect on the KDXD. However, the inhibition was not strong enough to explain the observed reduced activity and we had to lower the enzyme activity to 25% of the isolated enzyme. In addition, the KGSADH was inhibited by NADH, α-ketoglutarate and KDX (Supplementary Fig. 8), and a synergistic effect of KDX on NADH inhibition was observed (Supplementary Table 7). Thus, for the simulation of the reference state one-pot cascade, we adapted one model parameter for the KDXD reaction and this finalized the model construction process. From here on, no model parameters are changed anymore and all model simulations are predictions.

**Model validation in one-pot cascade perturbation experiments**. In a first experiment for model validation, we introduced a NAD$^+$ recycling system into the one-pot cascade experiment. Model analyses suggested that the incomplete conversion of D-xylose to α-ketoglutarate in the complete pathway was due to NADH inhibition, which was tested by including an NAD$^+$ recycling reaction to the system (Supplementary Note 1). For this, we used the same protein concentrations as in the reference state, plus 10 U lactate dehydrogenase and 15 mM pyruvate, ensuring that NADH produced by XDH and KGSADH was efficiently re-oxidized, resulting in consistent high NAD$^+$ concentrations and leading to lactate formation reflecting the dehydrogenases activity. In the presence of NAD$^+$ recycling, D-xylose was indeed completely converted into α-ketoglutarate (98% after 500 min and 84% after 350 min; Fig. 3b, compared with 38% after 350 min without NAD$^+$ recycling; Fig. 3a). After including the NAD$^+$ recycling system in the kinetic model, it was able to precisely predict the pathway dynamics and the model prediction that NAD$^+$ recycling would strongly enhance the pathway flux was confirmed.

In the second experiment for model validation, we addressed the significance of the XLA for the Weimberg pathway. The second enzyme in the pathway, XLA, seems not to be essential for the pathway, as D-xylonolactone is converted to D-xylonate in a non-enzyme catalysed reaction. In engineering projects, the XLA is often omitted, as it is not essential[19], and its omission would also prevent accumulation of D-xylonate, which was shown to be toxic in *E. coli*, *C. glutamicum* and yeast[19,21–26]. To test whether XLA has an effect on the pathway flux, we omitted the enzyme in the reference state incubation while keeping the other four enzymes at the reference state concentrations (Supplementary Note 1). As shown in Fig. 3c, the model from the reference state simulation could predict the system in absence of XLA. Strikingly, although a much higher D-xylonolactone concentration was observed compared with the reference state, the overall production rate of α-ketoglutarate was not much different from the one in the presence of XLA (Fig. 3a). This demonstrates that under the chosen assay conditions, the non-catalysed conversion of D-xylonolactone to D-xylonate does not become limiting and was sufficient for the low α-ketoglutarate production rates (but see below for optimized pathway conditions).

**Model application for rational pathway design**. With the final model, computational design for pathway optimization can now be performed to identify optimal enzyme ratios for specific applications or to test specific pathway characteristics (Supplementary Note 1 and Supplementary Table 8).

First, we used computational pathway design to optimize the in-vitro enzyme cascade for highest conversion efficiency. With the kinetic model, we analysed the optimal protein ratios for the fastest conversion rates of D-xylose to α-ketoglutarate with a given total protein amount (22.5 μg ml$^{-1}$, the same as in the other one-pot cascade experiments). The computational analyses were performed in the absence and presence of NAD$^+$ recycling (Supplementary Table 8). As shown in Fig. 3b, for the reference state analysis with NAD$^+$ recycling, 80% of the D-xylose was converted to α-ketoglutarate in 350 min; however, when we distribute the protein amount over the different enzymes in the pathway for the fastest conversion (with NAD$^+$ recycling; Supplementary Table 8), the model simulation predicted a full conversion within 100 min, which was in good agreement with experimental results derived from fresh as well as stored enzyme preparations with altered enzyme activities (Fig. 3d and Supplementary Table 9). The biggest change in enzyme concentrations between the reference state and the optimized state are the relative concentrations of KDXD and KGSADH, where a shift to KDXD speeds up the conversion of KDX, which accumulates in the reference state. In contrast, when we analysed the optimal enzyme distributions for the pathway in the absence of NAD$^+$ recycling (Supplementary Table 8), KGSADH is by far the most abundant enzyme. Under these conditions, only 64% conversion is obtained in 600 min. This demonstrates that the model enables rational pathway design and can be easily adopted to different enzyme preparations or different assay conditions.

We also performed computational design in the absence of XLA to test whether the non-enzymatic conversion of D-xylonolactone becomes limiting at the higher pathway fluxes under optimal protein distribution (Supplementary Table 8). Notably in the optimized state, the model predicts that the XLA becomes limiting for maximal pathway flux. Now much more protein is shifted to the first enzyme (XDH) for the optimal protein distribution over the pathway enzymes and the KGSADH is the least abundant enzyme in the pathway (Supplementary Table 8).

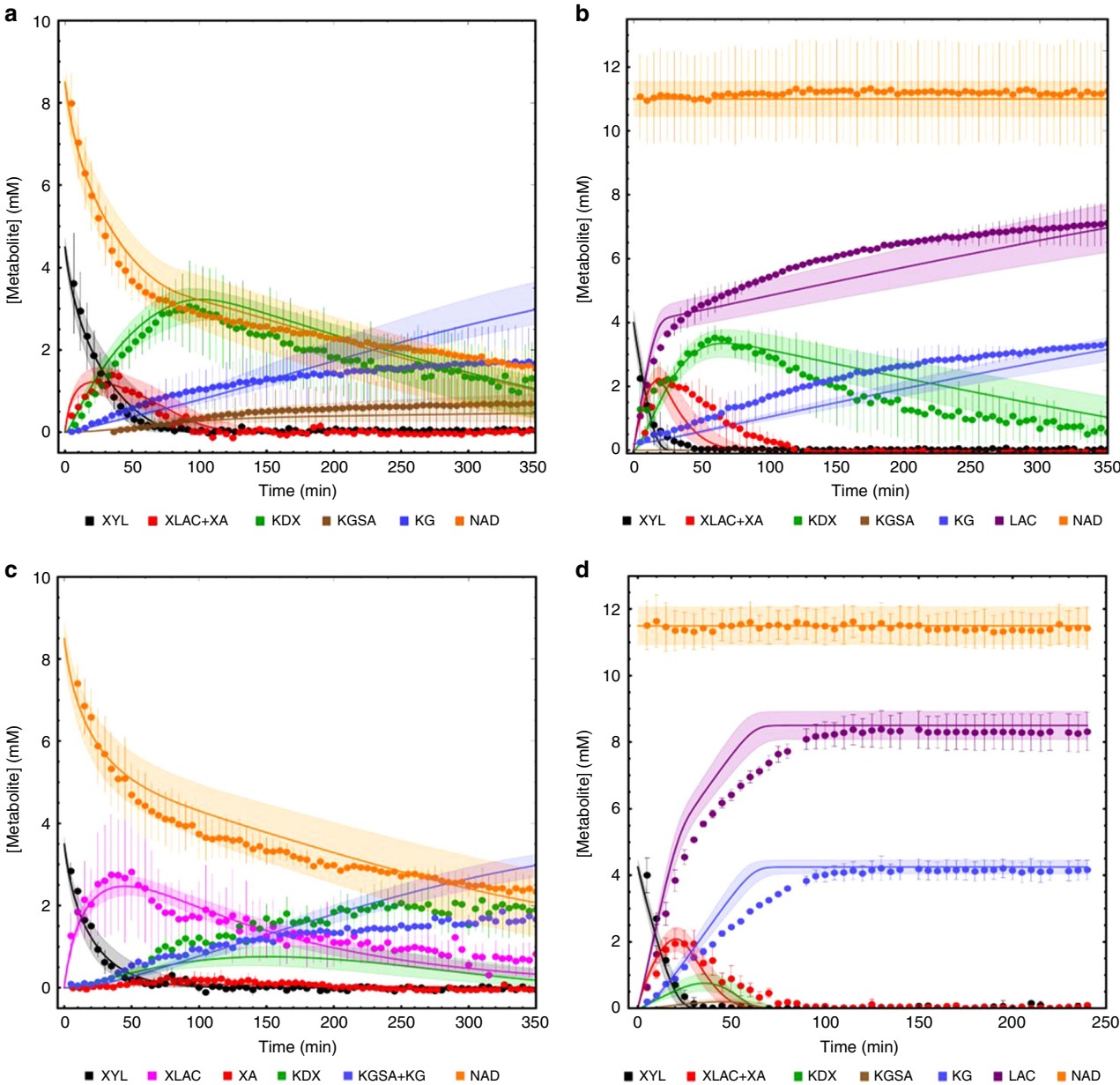

**Fig. 3 Complete pathway analysis.** The complete pathway analysis (one-pot cascade) with all enzymes present at $t = 0$ min. Metabolite concentrations were determined using NMR and the experiments were simulated with a detailed mathematical model (see Supplementary Note 1 for the model description). The results for the reference state incubation (without NAD⁺ recycling) and the effect of NAD⁺ recycling using LDH and pyruvate is shown for the reference state in **a** and **b**, respectively. In **c**, the XLA was not added to the incubation, to test whether the non-enzymatic conversion of D-xylonolactone to D-xylonate limits the overall pathway flux, and in **d** the relative enzyme concentrations were changed (with the same total enzyme concentrations) based on model simulations for the optimal conversion, i.e., highest conversion efficiency to α-ketoglutarate. The number of independent experiments equals $n = 2$ (Fig. 3a, b and c) using two independent isolates for all panels. In the experiment of **d**, enzyme preparations from two independent enzyme expressions and purifications were used either as fresh enzymes (duplicate) or as stored enzymes with reduced activity (at adapted concentrations, see Supplementary Note 1 and Supplementary Table 9) ($n = 3$). Error bars represent the SEM for the independent experiments and the shaded bands indicate the simulation confidence showing the solution space for the model simulations with a 10% error margin for the parameters. For abbreviations, see Fig. 1; LAC, lactate. Source data are provided as a Source Data file. It also can be accessed via the following DOI and SED-ML: Fig. 3a: https://doi.org/10.15490/FAIRDOMHUB.1.ASSAY.987.2 or https://jjj.bio.vu.nl/models/experiments/shen2020_fig3a/simulate; Fig. 3b: https://doi.org/10.15490/FAIRDOMHUB.1.ASSAY.989.2 or https://jjj.bio.vu.nl/models/experiments/shen2020_fig3b/simulate; Fig. 3c: https://doi.org/10.15490/FAIRDOMHUB.1.ASSAY.988.2 or https://jjj.bio.vu.nl/models/experiments/shen2020_fig3c/simulate; Fig. 3d: https://doi.org/10.15490/FAIRDOMHUB.1.ASSAY.991.2 or https://jjj.bio.vu.nl/models/experiments/shen2020_fig3d/simulate.

Next, we studied whether our model is only applicable for cascade reactions with purified enzymes or whether it can also be useful in vivo for analyses of cell-free extract incubations. For this, we first determined the activity of the Weimberg enzymes in cell-free extracts of *Caulobacter* cells grown on either D-glucose or

D-xylose (Supplementary Note 1, Supplementary Fig. 12 and Supplementary Table 10). In accordance with previous transcriptome studies[43], high activities for the complete set of enzymes were only observed in D-xylose-grown cells, whereas in D-glucose-grown cells only high XLA activity was observed

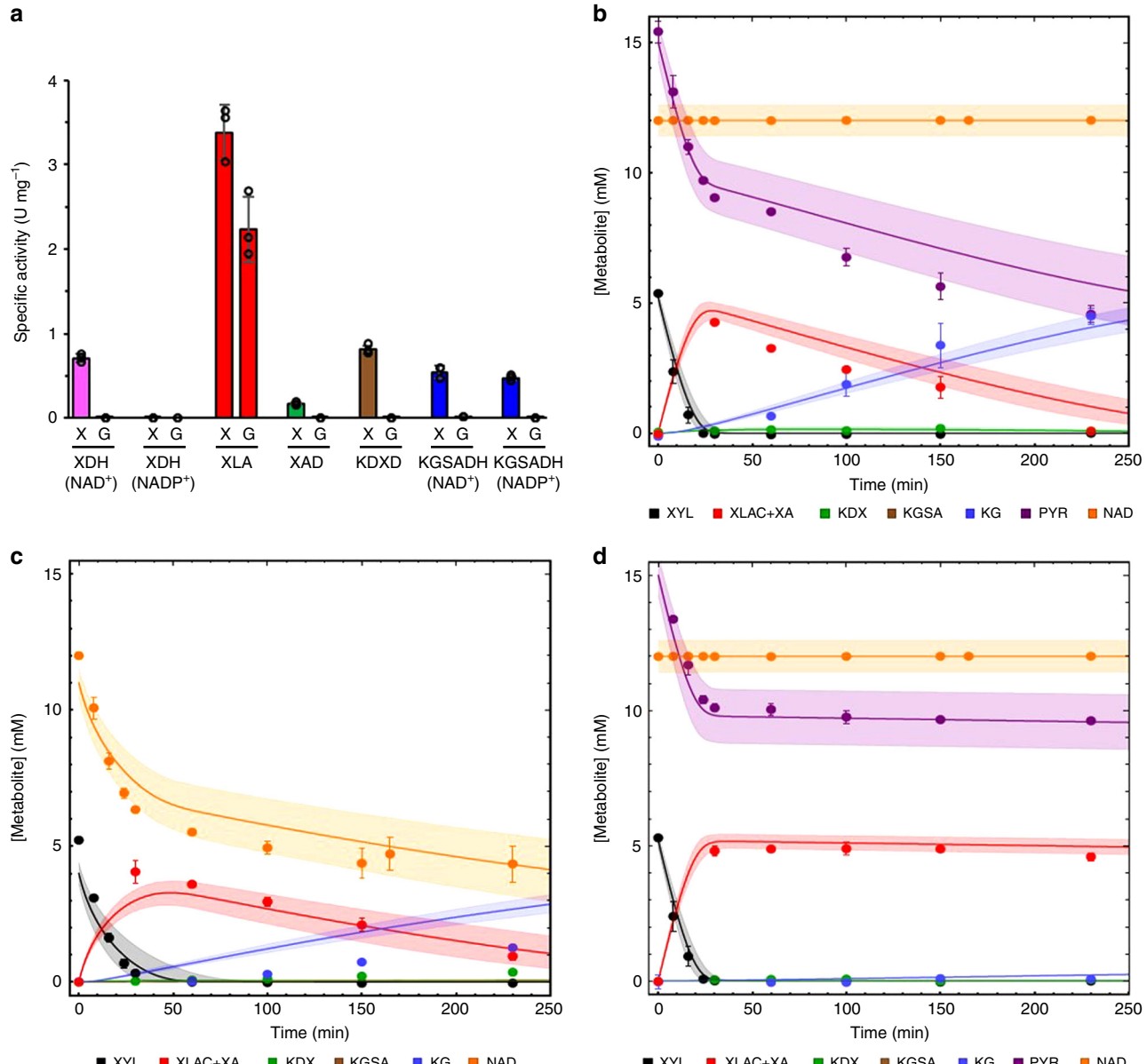

**Fig. 4 Enzyme activities in *C. crescentus* NA1000 cell-free extract and conversion rates.** *C. crescentus* NA1000 was grown on 0.2% (w/v) D-xylose (X) or D-glucose (G) to exponential phase (Supplementary Fig. 12) and Weimberg enzyme activities were determined in cell-free extracts (**a**). The specific activity of XDH and KGSADH was assayed using NAD+ or NADP+ as cofactor. All activity measurements in **a** were performed in triplicate (*n* = 3), mean values are shown and the error bars indicate the standard deviation (SD) (Supplementary Table 11). The conversion rates in cell-free extracts were determined in the presence of NAD+ recycling with Mn2+ (**b**), in the absence of NAD+ recycling with Mn2+ (**c**) and in the presence of NAD+ recycling without Mn2+ (**d**). Metabolites were determined enzymatically after protein precipitation (for details, see Supplementary Note 1). Error bars represent the SEM for the two independent experiments (*n* = 2) and the shaded bands show the solution space for the model simulations with a 10% error margin for the parameters. For abbreviations, see Fig. 1; PYR, pyruvate. Source data are provided as a Source Data file. It also can be accessed via the following DOI and SED-ML: Fig. 4b: https://doi.org/10.15490/FAIRDOMHUB.1.ASSAY.993.2 or https://jjj.bio.vu.nl/models/experiments/shen2020_fig4b/simulate; Fig. 4c: https://doi.org/10.15490/FAIRDOMHUB.1.ASSAY.994.2 or https://jjj.bio.vu.nl/models/experiments/shen2020_fig4c/simulate; Fig. 4d: https://doi.org/10.15490/FAIRDOMHUB.1.ASSAY.995.2 or https://jjj.bio.vu.nl/models/experiments/shen2020_fig4d/simulate.

(Fig. 4a and see Supplementary Table 11 for more detail). The protein concentration for each of the Weimberg enzymes in the cell-free extract was calculated from the specific activity in the extract (U mg⁻¹ total protein) and the specific activity of the purified enzyme (U mg⁻¹ enzyme).

Reflecting the high activities, we observed a high rate for D-xylose conversion to α-ketoglutarate in the cell extracts. Due to the complex mixture of the extracts, we could not use the NMR for

metabolite determinations in these experiments and used enzyme-based quantification methods in off-line analyses of samples (Supplementary Table 12), where enzyme activities were rapidly stopped by protein denaturation. Despite the complex enzyme mixture, the model predicted the pathway dynamics in the cell-free extract incubations quite accurately (Fig. 4), which is remarkable given that we simulate an experimental set-up with thousands of enzymes with a model based on five kinetic rate equations.

To test our hypothesis that XAD and KGSADH are limiting reactions in the cell-free extracts, we determined the sensitivity of the Weimberg pathway flux for $Mn^{2+}$ supplementation (0.15 mM addition) and $NAD^+$ recycling. As shown in Fig. 4, only in the presence of $NAD^+$ recycling and $Mn^{2+}$ supplementation, full conversion of 5 mM D-xylose to α-ketoglutarate is observed (Fig. 4b). Without $NAD^+$ recycling (but with $Mn^{2+}$ added), a significant reduction in the conversion rate was observed and only 1.3 mM α-ketoglutarate is formed in the incubation (Fig. 4c). Finally, if no additional $Mn^{2+}$ is added to the incubation, even in the presence of $NAD^+$ recycling, no α-ketoglutarate is formed and the reaction stalls at the level of D-xylonate (Fig. 4d). Thus, also in cell-free extracts, extra $Mn^{2+}$ must be added for XAD activity and $NAD^+$ recycling is crucial for full conversion of D-xylose to α-ketoglutarate.

Our studies demonstrate that the model was useful for the analysis of cell-free extract incubations. The implementation of the model was straightforward and required only specific activity measurements of the enzymes in the cell extract. The iterations between experiment and model allowed for a quick identification of the limiting reactions and our analyses point at XAD and XDH/KGSADH as sensitive reactions, due to their dependence on metal ions and $NAD^+$ regeneration, respectively.

**Biotechnology.** Finally, we addressed how the engineered in-vitro enzyme cascade can be used for added-value product formation. As experienced also in this study, many enzymatic analyses are hampered by the commercial availability and costs of the respective substrates. Thus, KDX and KGSA had to be produced using the in-house established enzyme cascade. As shown in Fig. 2c, all enzyme reactions in the Weimberg pathway run to completion and thus can be used for efficient product formation from cheap substrates.

To demonstrate the application potential, we tested the production of 2-keto-3-deoxy sugar acids by the XAD of *C. crescensus* at an industrially relevant scale. As shown previously for the *Thermoproteus tenax* gluconate dehydratase[44], the dehydratase reaction runs to completion and allows for the formation of stereospecific pure 2-keto-3-deoxy-D-gluconate (KDG). A detailed characterization of the *C. crescensus* XAD revealed a broad substrate specificity, accepting D-xylonate, D-galactonate and D-gluconate as substrates (Supplementary Table 2). We used the enzyme for the large-scale production of KDX, KDG and 2-keto-3-deoxy-D-galactonate. From 2.5 g D-gluconic acid, in 3 days, 2.3 g of KDG lithium salt were produced; similarly, 2.1 g D-xylono-1,4-lactone was fully converted to KDX within a day and 500 mg D-galactonic acid lactone was converted in 2 days to 400 mg of 2-keto-3-deoxy-D-galactonate lithium salt (for details, see Supplementary Note 3).

A final example for the application is the production of 4-hydroxyisoleucine, which is a very promising dietary supplement in the treatment and prevention of type II diabetes[45]. The L-isoleucine dioxygenase from *Bacillus thuringiensis* (BtDO) was previously reported to catalyse the conversion of different amino acids including L-isoleucine to the corresponding hydroxyl amino acids[45,46], using α-ketoglutarate as a reductant. By combination of the *C. crescentus* Weimberg enzyme cascade with the dioxygenase from *B. thuringiensis*[47], 2 mM L-isoleucine was completely converted to (2S, 3R, 4S)-4-hydroxyisoleucine by using 3 mM D-xylose after 3 h (for details, see Supplementary Note 4 and Supplementary Fig. 28).

## Discussion

Many metabolic engineering and synthetic biology projects are hampered by incomplete biochemical knowledge of the enzymes in the metabolic systems[28,29,48]. Due to a lack of quantitative models, rational pathway design is not widely applied and time-consuming combinatorial approaches are used instead. For the Weimberg pathway of *C. crescentus*, only the XDH, the XAD and the KDXD had been partially kinetically characterized[38–42], and although XLAs were characterized, e.g., from *Azospirillum brasilense*, *Haloferax volcanii* and partially from *C. crescentus*[9,10,33,49], kinetic constants for *C. crescentus* were not available. As a prerequisite for rational pathway design and detailed kinetic modelling, we herein established purification procedures and kinetic enzyme assays, and in addition developed protocols for effective synthesis and determination of pathway intermediates. All five Weimberg pathway enzymes were characterized in detail in initial rate experiments. These served as the starting point for a number of iterations between model and experiment to construct and validate a quantitative mathematical model for the Weimberg pathway (Fig. 5). The combination of experimental approaches and the integration with the modelling workflow established here can be applied generically to metabolic pathways. The method is elegant and clearly separates model construction, validation and application, and in combination with FAIR data management[50] leads to a transparent and reproducible modelling approach.

With our initial rate experiments, we confirmed the results for the XDH, XLA and XAD previously determined[33,38–41]. We observed some variations in kinetic constants, which are most likely due to differences in the assays that were used (e.g., with respect to buffer system, temperature and metal ions) and differences in enzyme-storage conditions[42]. Importantly, we included in our analyses also product inhibition and allosteric effectors, essential for the construction of the mathematical model.

Our results demonstrate that the efficient recycling of $NAD^+$ and the addition of divalent metal ions (we showed $Mn^{2+}$ was the most effective) are crucial for pathway performance in the enzyme cascade and in cell-free extract assays, pointing at XAD and XDH/KGSADH as steps in the pathway that are sensitive to experimental conditions. Flux limitation via XAD was recently suggested in a combinatorial approach with a mixed in-vitro enzyme cascade, and this limitation has been attributed to insufficient FeS cluster building essential for dehydratases of the ILVD (isoleucine valine biosynthesis dehydratase)/EDD (Entner-Doudoroff dehydratase) enzyme family[33,51]. Therefore, in metabolic engineering approaches in yeast that suffer from low XAD activity, either the iron uptake was improved[27] or the *FRA2* gene encoding an iron regulon repressor was deleted to induce FeS metabolism and thus improve XAD performance[26,51]. In their optimized yeast strain ((Δ*fra2*, *XylB*, 4x(*xylD*, *xylX* and *ksaD*))[26], the authors observed low D-xylose consumption with D-xylonate excretion leading to acidification of the growth medium in shake flasks and oxygen deficiency was suggested as a possible reason for low efficiency of the oxidative Weimberg pathway. In bioreactor experiments, the accumulation of KGSA rather than KDX was detected; this together with the D-xylonate accumulation (see above) point at KGSADH and XAD as rate-controlling steps under these conditions[26]. These results are in line with our observation where we pointed at these two reactions as sensitive steps and with our metabolic control analyses (MCA) (see below) where we found control in the dehydrogenases (at high NADH levels) or XAD (at lower NADH levels).

The significance of the XLA in the Weimberg pathway was studied using the Weimberg model and we could show that at low pathway flux the non-enzymatic hydrolysis might be sufficient, however, at higher pathway fluxes, it becomes limiting. The significance of the lactonase was indicated before in an in-vitro combinatorial approach[33]; however, the kinetic characterization and modelling used in the current study enable a quantitative

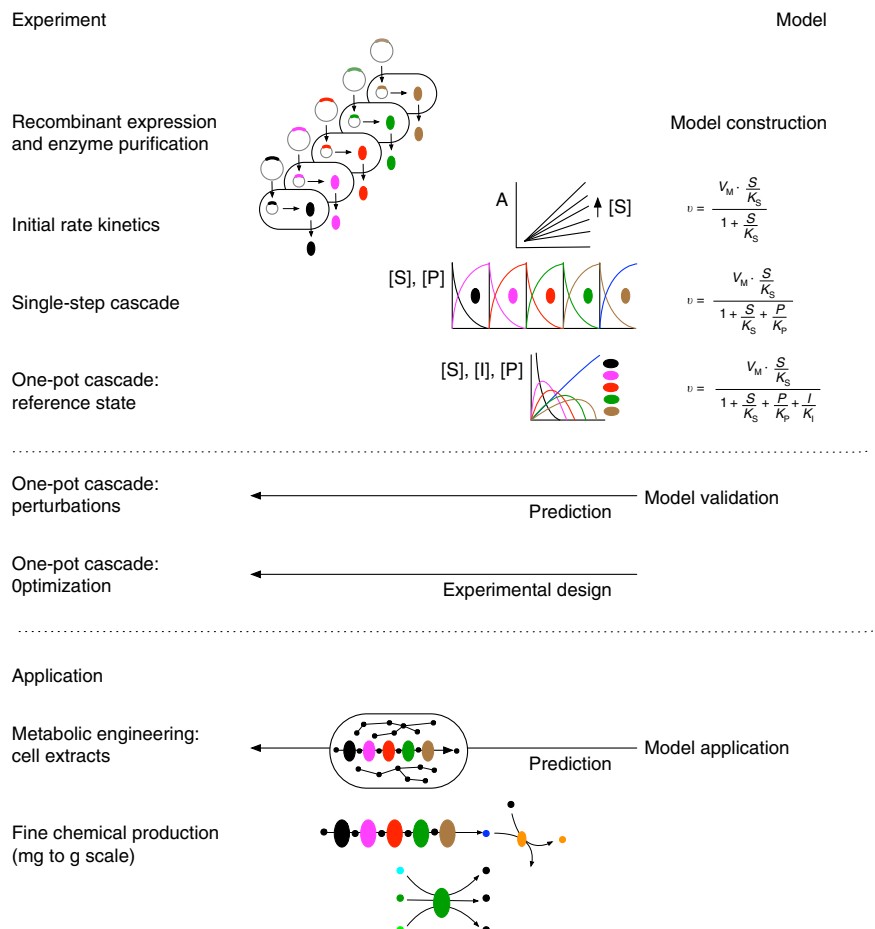

**Fig. 5 Strategy for computational pathway design using an iterative experimental and modelling approach.** The methodology was established for the *C. crescentus* Weimberg pathway, a five-step pathway for conversion of D-xylose to α-ketoglutarate.

determination of the contribution of lactonase to the pathway flux. In *C. crescentus*, all five genes of the Weimberg pathway including the XLA have been shown to be essential for growth on D-xylose[13]. Interestingly, in metabolic engineering approaches in yeast and *C. glutamicum*, the omission of the XLA gene was beneficial, which could be related to a decrease in D-xylonate accumulation and toxification problems. D-xylonate accumulation is also discussed to be the underlying mechanism leading to loss of XDH in yeast in an adaptive LAE approach to improve D-xylose utilization[26]. Finally, in yeast the increase of the gene dosage of the last three enzymes and the insertion of an alternative KGSADH identified in *C. glutamicum* (*ksaD*, Cg0535)[23] was beneficial ((Δ*fra2*, *XylB*, 4x(*xylD*, *xylX* and *ksaD*)) to establish a functional Weimberg pathway[26]. Reductionist approaches in *C. glutamicum* and in *P. putida* revealed that only the addition of one of the two dehydratases (XAD or KDXD) or of XAD, respectively, is required to enable growth on D-xylose as the sole energy and carbon source due to the presence of hitherto unknown endogenous enzymes[20,23]. With the available Weimberg model, simple, specific activity measurements in cell-free extracts in combination with quantitative model analyses could significantly improve metabolic engineering approaches and provide help for strain design including cloning strategies and strain optimization, e.g., in respect to the required gene dosage to prevent imbalance of the pathway reactions leading to intermediate accumulation.

For identification of rate-limiting steps in metabolic pathways, the catalytic efficiency (defined as kcat/Km) is often used. For the

Weimberg pathway, Tai et al.[42] found that the KDXD had the lowest catalytic efficiency and postulated this as the rate-limiting step. For our kinetic parameters, the XAD had the lowest catalytic efficiency and we had a notably higher kcat value for the KDXD, possibly due to differences in the kinetic assay and in enzyme-storage conditions. However, we would like to stress the limitations of using the catalytic efficiency to identify rate-limiting steps. The kcat/Km value gives an indication of the activity per enzyme at low substrate concentrations and does not take the expression level of the enzyme into account. Clearly, if an enzyme with low catalytic efficiency is sufficiently high expressed, it will not be rate limiting. In the end, a complete quantitative model is the ultimate tool to identify rate-limiting steps, as it integrates all kinetic parameters and is not limited to low substrate concentrations. A suitable analysis method for quantifying the flux control of the individual enzymes in a system is metabolic control analysis (MCA, for reviews, see refs. [52,53]). We clamped the concentrations of D-xylose, α-ketoglutarate and NAD(H) in the cell-free extract model so that the system could reach a steady state, and performed MCA on this model. We observed that at high NADH concentration, flux control resides in the dehydrogenases, notably in the XDH, and lowering the NADH concentration leads to a control shift to the XAD. At low NADH concentration, the system did not reach a steady state due to accumulation of D-xylonate and KDX, indicative of rate limitations for the dehydratase reactions. These results reflect many of the metabolic engineering issues encountered for the Weimberg pathway, which can now be analysed with the kinetic model. For

optimal performance of the pathway, a low NADH/NAD$^+$ ratio is important (good aeration) and care must be taken that the hydratases have sufficient activity (expression level and metal ion concentrations (Mn$^{2+}$ or Mg$^{2+}$) should be high enough), to balance the production of D-xylonate and KDX. At low pathway fluxes, the XLA is not essential and its omission can be advantageous to prevent D-xylonate accumulation, but this will lead to a bottleneck at higher flux values.

In conclusion, we have illustrated a generic model construction and validation approach, by starting with initial rate kinetics to build parameterized rate equation for substrate dependencies; in a first validation, we showed in conversion assays that product inhibition was significant for most of the reactions and needed to be included in the model. In the second round of validation, in one-pot cascades we observed allosteric regulation of some of the enzymes, which was added to the model after confirmation in initial rate kinetic experiments and calibrated on the one-pot reference state experiment. The final resulting model was validated by testing its prediction of pathway dynamics with NAD$^+$ recycling and the effect of omission of XLA. As a first model application, we analysed the model for optimal protein distribution over the enzymes for the highest production rate to α-ketoglutarate and later applied our model to cell-free extracts of *C. crescentus*. Finally, we demonstrated that the XAD and the Weimberg enzyme cascade is applicable for biotechnological applications.

## Methods

**Gene cloning and protein overexpression.** The five genes of the D-xylose operon, i.e., *CCNA_00862–00866*, were amplified by PCR using gene-specific primers (Supplementary Table 1) and the genomic DNA of *C. crescentus* NA1000 as template. The PCR products were cloned into pET (Novagen) expression vectors (Supplementary Table 1) and successful cloning was confirmed by sequencing (LGC genomics, Berlin). For expression, the respective plasmids were transformed into the *E. coli* BL21-pRIL (DE3) or Rosetta (DE3, Stratagene) (Supplementary Table 1). Overexpression was performed in LB medium containing the appropriate antibiotic. The cells were grown at 37 °C to an OD$_{600}$ of 0.6–0.7 and induced with 1 mM isopropyl-β-D-thiogalactopyranoside. After induction, the cultures were further incubated at 37 °C for 4 h or at 30 °C or at 21 °C for 18–22 h (Supplementary Table 1), respectively. Cells were collected by centrifugation (15 min, 8630 × *g*, 4 °C) and the pellets were stored at −70 °C. Due to the choice of the vectors, all of the *C. crescentus* enzymes were N-terminally histidine tagged. The recombinant L-isoleucine dioxygenase from *B. thuringiensis* (BtDO) contained a C-terminal 6× histidine tag[47].

**Protein purification.** For XAD, XLA, KGSADH and BtDO, all buffers were supplemented with 1 mM of dithiothreitol. The cell pellet was resuspended in LEW buffer (Macherey-Nagel) and disrupted by sonication on ice for 3 × 5 min (1 min interval in between). Cell debris was removed by centrifugation (45 min, 21,130 × *g*, 4 °C) and the histidine-tagged proteins were purified from the supernatant by immobilized metal ion affinity chromatography using Ni-TED (nickel (tris(carboxymethyl)ethylene diamine) columns (Macherey-Nagel) according to the manufacturer's instructions. The elution fractions containing the respective recombinant proteins were collected and concentrated (Vivaspin®, Satorius Stedium Biotech). For the *C. crescentus* proteins, the concentrated samples were separated via a size-exclusion chromatography column (HiLoad 26/60 Superdex 200 prep grade, Amersham Biosciences) pre-equilibrated with 50 mM HEPES/NaOH (pH 7.3, room temperature) containing 300 mM NaCl. XAD was immediately supplemented with MnCl$_2$ after purification (17 μmol MnCl$_2$ per mg protein) and is then stored at −70 °C. The purified XLA and KDXD were stored at 4 °C, whereas XDH and KGSADH were stored at −70 °C. BtDO was washed three times after Ni-TED purification with 10 ml of 50 mM HEPES/NaOH (pH 7.3, RT) using a 10 kDa cut-off centrifugal concentrator (Vivaspin® 20, Satorius Stedium Biotech) and was finally concentrated to 5–10 mg ml$^{-1}$ and stored at 4 °C until use. The protein concentration was determined using the Bradford assay with bovine serum albumin (Merck) as standard.

**Growth of *C. crescentus* NA1000.** *C. crescentus* NA1000 was grown in M2 minimal salts medium (6.1 mM Na$_2$HPO$_4$, 3.9 mM KH$_2$PO$_4$, 9.3 mM NH$_4$Cl, 0.5 mM MgSO$_4$, 0.5 mM CaCl$_2$ and 10 μM FeSO$_4$-EDTA chelate solution) supplemented with 0.2% (w/v) D-xylose or D-glucose[43], and the optical density at 600 nm (OD$_{600}$) was monitored. For activity tests, cells were collected in the exponential phase (OD$_{600}$ around 1) by centrifugation (15 min, 8150 × *g*, 4 °C) and

the cells were frozen at −70 °C until use. The residual D-xylose concentration in the culture supernatants after centrifugation (10 min, 21,130 × *g*) was determined as described in Supplementary Table 12.

**Preparation of enzyme substrates.** D-xylonate and D-galactonate were prepared from D-xylonic acid-1,4-lactone (Carbosynth, UK) and D-galactonic acid-1,4-lactone (Sigma-Aldrich), respectively. The respective sugar acid lactones (1 M) in H$_2$O were incubated with the same volume of 1.2 M NaOH at 37 °C for 1 h. The mixture was afterwards diluted with four volumes of 50 mM HEPES/NaOH (pH 7, RT). The obtained 0.1 M D-xylonate/D-galactonate solutions were either used directly or stored at −20 °C.

KDX, KDG and 2-keto-3-deoxy-D-galactonate were produced enzymatically using the purified *C. crescentus* XAD from D-xylonate, D-gluconate and D-galactonate as row materials, respectively. XAD (3 U ml$^{-1}$; stored in the presence of 17 μmol MnCl$_2$ per mg protein), 30 mM D-xylonate, D-gluconate, or 3 mM D-galactonate were incubated in 50 mM HEPES/NaOH (pH 7.5, 37 °C) at 37 °C for 2 h. Proteins were afterwards removed by a 10 kDa cut-off centrifugal concentrator (Vivaspin® 6, Satorius Stedium Biotech). KGSA (30 mM) was prepared from D-xylonate by the same method using 3 U ml$^{-1}$ of each *C. crescentus* XAD and KDXD. All of the prepared compounds were verified by $^1$H-NMR and $^{13}$C-NMR at a 600 MHz Bruker spectrometer using H$_2$O/D$_2$O (9:1, v/v) as solvent at 37 °C.

**Enzyme assays.** All standard enzyme assays were performed in 100 mM HEPES/NaOH (pH 7.5, 37 °C) at 37 °C. Reaction mixtures (without substrate) were pre-incubated at 37 °C for 2 min and the reaction was started by the addition of substrate. Every assay was carried out in triplicate.

XDH activity was determined in a continuous assay (total volume 500 μl) by following the formation of NADH as the increase in absorbance at 340 nm, using D-xylose (0.3–5 mM) or L-arabinose (5–300 mM) as substrate, NAD$^+$ (0.01–3 mM) as cofactor, and 136 ng purified enzyme. For substrate specificity testing, 30 mM D-glucose, D-galactose, L-xylose, D-arabinose or D-ribose with 5 mM NAD$^+$ or 1 mM NADP$^+$ were used.

XLA activity was measured at 340 nm by coupling the formation of D-xylonate from D-xylonic acid-1,4-lactone to the reduction of NAD$^+$ via the *C. crescentus* enzymes XAD, KDXD and KGSADH. The standard assay (total volume 500 μl) contained 6.9 ng XLA, 0.88 U XAD, 0.52 U KDXD, 0.34 U KGSADH, 5 mM NAD$^+$ and 0.15-2 mM D-xylonic acid-1,4-lactone.

XAD activity was determined by the thiobarbituric acid assay[54] in a discontinuous assay using D-xylonate (0.15–10 mM), D-gluconate (0.25–12.5 mM) or D-galactonate (0.25–12.5 mM) as substrates and 3.28 μg of XAD (stored in the presence of 17 μmol MnCl$_2$ per mg protein). The assay was performed in a total volume of 1 ml. Samples (100 μl) were taken at different time points and the reaction was stopped by the addition of 10 μl 12% (v/v) trichloroacetic acid and samples were transferred on ice for 10 min. Precipitated proteins were removed by centrifugation at 21,130 × *g*, 4 °C for 15 min. To 50 μl of the obtained supernatant, 125 μl of 25 mM periodic acid (dissolved in 0.25 M H$_2$SO$_4$) was added and the mixture was incubated at room temperature for 20 min. The oxidation reaction was terminated by adding 250 μl of 2% (w/v) sodium arsenite (dissolved in 0.5 M HCl), followed by addition of 1 ml of 0.3% (w/v) thiobarbituric acid and heating at 100 °C for 10 min to develop the chromophore. The obtained solution was mixed with an equal volume of dimethylsulfoxide (DMSO) and the absorbance was determined at 550 nm. When the absorbance exceeded 1, the sample (before addition of DMSO) was diluted with water. The assay system was calibrated using the respective 2-keto-3-deoxy sugar acids (verified by NMR) in concentration between 0 and 1.5 mM. The extinction coefficients of KDX, KDG and 2-keto-3-deoxy-D-galactonate were determined to be 80.95 mM$^{-1}$ cm$^{-1}$, 110.95 mM$^{-1}$ cm$^{-1}$ and 62.57 mM$^{-1}$ cm$^{-1}$, respectively.

KDXD activity was assayed in a continuous coupled assay with KGSADH as auxiliary enzyme using KDX as substrate. The standard assay (total volume 500 μl) contained 0.14 μg KDXD, 1.21 U KGSADH, 5 mM NAD$^+$ and 0.05–5 mM KDX. The activity was monitored as NADH formation at 340 nm.

KGSADH activity was determined in a continuous assay (total volume 500 μl) as NAD(P)$^+$ reduction at 340 nm, using KGSA (0.05–1 mM), succinic semialdehyde (0.05–7.5 mM) or glutaraldehyde (0.1–15 mM) as substrates and NAD$^+$ (0.01–12 mM) or NADP$^+$ (0.005–3 mM) as cofactor. The substrate specificity was tested with 5 mM D,L-glyceraldehyde, glycolaldehyde dimer, benzaldehyde, valeraldehyde, hexanal or isobutyraldehyde and 5 mM NAD$^+$/NADP$^+$.

The pH optima of all enzymes were determined in a mixed buffer system with MES, HEPES and TRIS (50 mM of each) in the range of pH 5.5–9.5 adjusted at 37 °C. The temperature optima were determined in 100 mM HEPES/NaOH (pH 7.5) at 25–75 °C. Thermostability was checked by incubating the enzyme in 100 mM HEPES/NaOH (pH 7.5) at 37 °C or 55 °C and samples were taken every 30 min till 3 h to test the residual activity under standard conditions.

**Enzyme activities in cell-free extracts.** The cell pellets of *C. crescentus* NA1000 grown on D-xylose and D-glucose, respectively, were resuspended in HEPES/NaOH buffer (pH 7.3, RT) supplemented with 300 mM NaCl and disrupted by sonication

on ice for 3 × 5 min (1 min interval in between). Cell debris was removed by centrifugation (45 min, 21,130 × g, 4 °C). The specific enzyme activities of the Weimberg enzymes in the cell-free extracts were determined photometrically as described in Supplementary Note 1 and Supplementary Table 10.

**Model construction and data management**. For the model construction, a bottom-up approach was used, starting with the kinetic characterization of the isolated enzymes, using initial rate kinetics. As each of the reactions ran to completion, we used irreversible kinetic rate equations of a Michaelis–Menten type but including product inhibition. The rate equations were fitted on the experimental data using the NonlinearModelFit function in Mathematica (www.wolfram.com). The Mathematica notebooks and experimental data are available on the FAIRDOMHub[55] (https://fairdomhub.org/investigations/284). The parameterization can be reproduced by downloading the data file and Mathematica notebook in the same directory and evaluate the notebook.

The next step in the model construction was to test the initial rate kinetics in its prediction of substrate conversion assays. As some of the enzymes were characterized in linked assays, it was not possible to determine product inhibition in the initial rate experiments. Thus, we fitted product binding constants using the conversion datasets (sequential and one-pot enzyme cascades) and the NMinimize function in Mathematica with a $\chi^2$-based objective function.

Subsequent model validations were made by predicting perturbation experiments such as cofactor regeneration, omission of the XLA enzyme (this reaction runs also without enzyme) and simulating a predicted optimal protein distribution for product formation. All model files are made available as SBML files. All data files are made available as RightField[56] annotated excel files. Model description files are made available as SED-ML[57] scripts on the FAIRDOMHub and JWS Online[58].

**Determination of intermediate concentration using NMR**. The conversion of D-xylose to α-ketoglutarate by purified proteins was monitored by [1]H -NMR and [13]C-NMR with a 600 MHz Bruker spectrometer using $H_2O/D_2O$ (9:1, v/v) as solvent at 37 °C (Supplementary Note 2, Supplementary Fig. 13–27 and Supplementary Table 13). The NMR spectra were recorded with data points every 5 min to follow the concentrations of the intermediates and products. Absolute concentrations were determined by integration in the [1]H NMR vs. the buffer signals. The reaction mixture (1 ml) contained 100 mM HEPES/NaOH (pH 7.5, 37 °C), 12 mM $NAD^+$ and 5 mM D-xylose-1-[13]C. It was pre-incubated at 37 °C for 2 min (without enzyme) and the reaction was started by the addition of enzyme(s). For the sequential enzyme cascade reaction, each reaction was run to completion before the next enzyme was added. For the one-pot reactions, proteins were mixed and the reaction was started by the addition of the enzyme mixture. The amount of each enzyme in the reactions is given in Supplementary Note 1. The one-pot cascade reactions were performed in the absence and presence of XLA and the effect of an $NAD^+$ recycling system was followed in the presence of 15 mM pyruvate and 30 U L-lactate dehydrogenase (LDH, Merck).

**Enzymatic quantification**. The conversion of D-xylose to α-ketoglutarate by the cell-free extracts was monitored by enzymatic quantification of the intermediate concentrations at different time points. The reaction mixture (5 ml) contained 2 mg cell-free extract proteins, 100 mM HEPES/NaOH (pH 7.5, 37 °C), 12 mM $NAD^+$, 5 mM D-xylose and either (1) 0.15 mM $MnCl_2$ and the $NAD^+$ recycling system as mentioned above; (2) 0.15 mM $MnCl_2$ without the $NAD^+$ recycling system or (3) only the $NAD^+$ recycling system without $MnCl_2$. The reaction mixtures were incubated at 37 °C in a water bath with constant stirring and samples were taken at 0, 8, 16, 24, 30, 60, 100, 150, 165 and 230 min for intermediate quantification. To determine the NADH concentration, 10 µl of these samples were transferred into 290 µl ice-cold water. These dilutions (200 µl) were transferred into microtest plates (ROTH) and the absorbance was determined at 340 nm using a microplate reader (Infinity 200 M, TECAN, Switzerland). To detect the concentration of the other intermediates, 150 µl of the samples were transferred to 15 µl 12% (v/v) tri-chloroacetic acid to stop all enzymatic reactions. The obtained samples were kept on ice for 10 min and the precipitated proteins were removed by centrifugation at 21,130 × g, 4 °C for 15 min. The resulting supernatants at different time points (i.e., α-ketoglutarate at 0, 60, 100, 150 and 230 min; pyruvate and D-xylose at 0, 8, 16, 24, 30, 60, 100, 150 and 230 min; D-xylonate, KDX and KGSA at 0, 30, 60, 100, 150 and 230 min) were used for the enzymatic quantification of intermediates. The concentration of α-ketoglutarate was determined using the α-ketoglutarate kit from Merck. The concentrations of the residual five Weimberg intermediates were linked to the consumption or production of NADH using auxiliary enzymes (for details see Supplementary Table 12). After 30 min incubation at 37 °C, 200 µl of each reaction assay was transferred into microtest plates to determine the absorbance at 340 nm using the microplate reader.

**Biotechnological application**. The detailed description for the application of the XAD for the production of 2-keto-3-deoxy sugar acids in g scale and the Weimberg enzyme cascade for (2S, 3R, 4S)-4-hydroxyisoleucine production is presented in the Supplementary Note 3 and 4, respectively.

**Reporting summary**. Further information on research design is available in the Nature Research Reporting Summary linked to this article.

## Data availability

Data supporting the findings of this work are available within the paper and its Supplementary Information files. A reporting summary for this Article is available as a Supplementary Information file. The datasets generated and analysed during the current study are available on the data and model management platform FAIRDOMHub[55] (https://fairdomhub.org/investigations/284), with DOI (doi:10.15490/FAIRDOMHUB.1.INVESTIGATION.284.2) or from the corresponding author upon request. The data structure on the FAIRDOMHub is an ISA structure (Investigation-Study-Assay). The above link is to the investigation level. There are four studies: (1) initial rate kinetics, (2) progress curves, (3) one-pot cascades, and (4) cell-free extract. Each of the studies has several assays. The initial rate kinetics has a data file (RightField[56] annotated excel file) for each of the five enzymes of the pathway (used in the Supplementary Figs. 4–8). The progress curves study has six assays each with a data file and two model files. The individual progress curves are used in Supplementary Figs. 9–11. The combined progress curves are shown in Fig. 2c in the main document. The one-pot cascade study has four assays. Each containing a data file and two model files (used in Fig. 3a–d). The cell-free extract study contains four assays. Three contain a data file and two model files (used in Fig. 4b–d). The last assay contains a model file for the steady-state analysis (see Discussion). The source data underlying Figs. 1c, 2c, 3, and 4, as well as Supplementary Figs. 1–12 and 28 are provided as a Source Data file.

## Code availability

All models and analysis scripts are made publicly available in the data and model management platform the FAIRDOMHub in an investigation, together with the experimental data (see above data availability statement for the description of the investigation, https://fairdomhub.org/investigations/284), with DOI (doi:10.15490/FAIRDOMHUB.1.INVESTIGATION.284.2). The models shen1, shen2 and shen3 are available as SBML files. All other analyses were conducted by Mathematica v12 (www.wolfram.com), which are available as notebooks in FAIRDOMHub. In addition, we created SED-ML[57] scripts to reproduce the figures in the manuscript for the default parameter values (i.e., excluding Monte Carlo simulations). All these scripts are available for online simulation at the JWS Online[58] platform using the following link (https://jjj.bio.vu.nl/models/experiments/?id=shen2020&model=).

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

## Acknowledgements

L.S., J.E., R.K. and B.S. acknowledge funding by MERCUR Pr-2013-0010. L.S., C.B., J.S. and B.S. acknowledge funding by the Federal Ministry of Education and Research (BMBF) grant HotSysAPP, 03120078A within the eBio (2) funding initiative. J.S. acknowledges funding from the DST/NRF, particularly for funding the SARCHI initiative (NRF-SARCHI-82813). We thank Dr Martin Thanbichler (Max Planck Institute for Terrestrial Microbiology, Marburg, Germany) for providng *C. crescentus* cultures.

## Author contributions

L.S. cloned and purified the *C. crescentus* Weimberg enzymes and performed all enzyme characterizations and cascade, and cell-free extract experiments. M.K., F.N. and J.N. performed NMR studies and data analysis. J.S. performed modelling and experimental design, and d.v.N. carried out the data management. R.M., B. Schönenberger and R.W. established the production of 2-keto-3-deoxy sugar acids by the XAD of *C. crescentus* at an industrially relevant scale. L.S., J.E. and R.K. established the production of 4-hydroxyisoleucine using the L-isoleucine dioxygenase from *B. thuringiensis* (BtDO) and the *C. crescentus* Weimberg cascade. C.B., J.S. and B.S. conceived the study and L.S., C.B., J.S. and B.S. contributed by writing the manuscript. All authors approved the manuscript.

## Competing interests

The authors declare no competing interests.
