## [Peer Review File · Nature Communications]

Reviewers' comments:

Reviewer #1 (Remarks to the Author):

In this paper the authors explore a potentially useful pathway for the breakdown of xylose (a major sugar from cellulose biomass): the Weimberg pathway, a 5 step pathway that converts xylose into a-ketoglutarate. Every step in the pathway is highly favorable thermodynamically at standard state and it is carbon conserving. But so far, the pathway has not worked well in engineered organisms. Here the authors carefully examine the pathway biochemically and find that one enzyme requires Mn^{2+} and other enzymes are inhibited by NADH. By introducing NADH recycling and Mn, they develop a well-functioning pathway in vitro. The lessons learned from a purified system could also be applied to a crude lysate system. There are definitely some very nice aspects of this work, particularly the characterization of the pathway first by basic enzyme kinetic assays, then by following conversions sequentially by NMR which was beautifully done. Yet I have some concerns about the model and what I view as odd choices in presentation.

Specific Comments:

1) The initial one pot reaction without NADH recycling didn't "go to completion." This is attributed to inhibition by various metabolites. Yet the inhibition results seen were quite modest and would not account for the reaction basically stopping. This explanation simply doesn't make sense to me. In fact, the authors had to arbitrarily decrease the activity of KGSADH in their model by a not insignificant 4-fold and increase the inhibition of NADH by a whopping 30-fold to fit the experiment. It seems to me there is something quite wrong with the model. I would like the authors to consider two possibilities:

a) A synergistic effect of several metabolites. What happens when you mix them together? Are the effects magnified?

b) Or more likely...Is it possible that the inhibition by NADH is not really inhibition of the enzyme, but an equilibrium issue? As pointed out by the authors, under standard state conditions the reactions are highly favorable, but at the lower concentrations that exist in the one pot system it is possible that you are closer to equilibrium. If so, it would make sense that you can't push the reaction further as NADH builds up. An equilibrium issue would also be completely consistent with the fact that NADH recycling makes everything work nicely.

2) Line 208: "It is unclear why the NADH inhibition is much stronger in the complete pathway analysis, compared to the individual conversion assay and the initial rate kinetics." I don't think you can say that NADH inhibition is stronger in the complete pathway, only that your MODEL requires it. Other than the fact that it makes the data fit better, there isn't really evidence that that's going on. In fact, your experimental data would suggest otherwise.

3) The authors organize the "Model Application for Computational Pathway Design" and the "Model Significance, Usability and Applications" section in a very confused manner in my opinion. They describe how they could computationally optimize the system to generate KG faster. That is not really a finding. You can generate many fantasies in a computer. What the reader wants to know next is does the model actually predict that result correctly, which would be great. But no, we have to go into a discussion of XLA. Then you claim the model has been "validated" starting on line 262, but it hasn't really happened at that point in the story. You have to wait until the next section to find out that in fact the model seems to do OKish. Throughout these sections it is very hard for the reader to know if you are talking about computational results or experimental results. I think this entire presentation needs to be rearranged.

4) The section on the Significance of XLA seems rather inane to me. The reaction can occur spontaneously which is nice to know, but the fact that when the other enzymes are fast enough the spontaneous reaction is rate limiting is simply obvious and doesn't require a computational model. I think this section should be deleted and replaced with a one sentence summary, if that.

5) I'm impressed that the predicted optimization by the model was reflected pretty well in the experiment of Fig. 3D. THIS is a nice validation of the model! Yet the authors manage to confuse this point by talking about how they used old enzymes that were somewhat inactivated over time. This is not interesting. Enzymes die over time and some enzyme preps are more active than others so you need to adjust for their specific activity. We all know that. I don't understand why you would swallow the really nice result with what seems to me an inane tangent.

6) The results in Fig. 3D are very strange. At 100 min, there is apparently complete conversion to KG, yet there's still 1 mM KDX present, which slowly disappears. Something doesn't add up. And where is it going?

7) The biotechnology section is really strange.

a) In one application the authors use a single enzyme from the Weimberg pathway to convert a handful of sugars. Using a single enzyme and producing gram quantities of material is as old as biochemistry. Why should anyone be impressed by that?

b) In the second application they say that they use the Weimberg pathway to generate KG, which can then be used as a sacrificial substrate in the production of hydroxyisoleucine. This seems to be more of an interesting application, but it was completely unclear what they did. They show a figure in the supplement with results, but there is no experimental description anywhere in the paper or the supplement. A supplementary figure showing the system set up and experimental details are essential if this is to be included in the paper.

8) I can't figure out what the different colored lines are in Suppl. Fig. 8d and what the vertical axis means.

9) I have no idea what Suppl. Fig. 12 is supposed to convey other than the fact that if you vary parameters in the model you get different results. There is no experimental data shown for comparison.

Minor things

10) The title should say more about what the paper is about or what the authors want to communicate. Maybe something like, "Using cell free modeling to identify and correct bottlenecks in the Weimberg Pathway for conversion of cellulosic biomass."?

11) I believe KGSADH can use BOTH NAD⁺ and NADP⁺? It's not clear why you include that in Fig. 1 since it is rather confusing when you discuss NADH inhibition for a reaction that seems to use NADP⁺ according to the figure. I understand that you want to be complete, but maybe you could state the fact that it's non-specific in the figure legend, but leave it as NAD⁺ in the actually figure? Or maybe it's just me.

Reviewer #2 (Remarks to the Author):

In the manuscript of Shen and coworkers the Weimberg pathway from *C. crescentus* was studied in detail by applying classical in vitro enzyme assays, sophisticated NMR analysis for intermediate dynamics and standard mechanistic pathway modeling. Novel results concern the inhibition patterns of selected enzymes by various pathway intermediates as well as the production of two new compounds using a single biocatalytic step or the extended Weimberg pathway, respectively. The topic is very interesting and the paper is well written. However, some of the claims are either not new or need further experimental support.

Major remarks:

- L85: Recently, several studies showed improved performance of the Weimberg pathway in prokaryotic and eukaryotic organisms (Ref. 1-4). These should be taken into account for discussion.
- When applying the one-pot cascade, the model fails to accurately simulate the measured trajectories for XLAC, XA (both Fig. 3A-D) and KDX (Fig. 3D). Please provide measures for goodness of fit (e.g. RMSE) for all model fits.
- What was the rationale for applying only simple Convenience kinetics to model each enzymatic step? A comparison of different kinetic approaches including Michaelis-Menten type and approximative formats would have substantially improved the modeling part.
- L234ff: For convincingly showing that the model is valid, the data from the first experiment with fresh enzyme and optimized abundances should have been included as Fig. 3D. The data from the 2nd experiment using long-term stored enzymes might be included in the Supplement. However, apart from KG, the model fits seem to be not very accurate and therefore the predictive power of the model is questionable.
- L309: No metabolite quenching was applied to immediately stop enzymatic conversion. Please justify.
- L320: Again, the model fits are not really accurate and proper statistics are missing!
- L424, L430: Recently, Brüsseler et al. could already prove experimentally that a lactonase is not required and, most importantly, not limiting for in vivo Weimberg pathway operation.

Minor remarks:

- The title is rather uninformative and should be more specific, e.g. “In vitro studies on enzymes from *Caulobacter crescentus* Weimberg pathway”
- Fig. 2C: The model fits for the XAD step are rather bad. The linear increase of KDX points to enzyme inhibition and the model already covers KDX and NADH as effectors. Please discuss.
- Fig. 2C: In one experiment there was an accumulation of KGSA over 6 mM, which is significantly higher than the initial 5 mM xylose. Please discuss.

References:

- 1) Radek, A. et al. (2017) Miniaturized and automated adaptive laboratory evolution: Evolving *Corynebacterium glutamicum* towards an improved D-xylose utilization. *Bioresour Technol* 245(Pt B):1377-1385
- 2) Brüsseler, C. et al. (2018) The myo-inositol/proton symporter IolT1 contributes to D-xylose uptake in *Corynebacterium glutamicum*. *Bioresour Technol*. 249:953-961
- 3) Brüsseler, C. et al. (2019) Alone at last! - Heterologous expression of a single gene is sufficient for establishing the five-step Weimberg pathway in *Corynebacterium glutamicum*. *Metab Eng Commun*. doi: 10.1016/j.mec.2019.e00090
- 4) Borgström, C. et al. (2019) Identification of modifications procuring growth on xylose in recombinant *Saccharomyces cerevisiae* strains carrying the Weimberg pathway. *Metab Eng*. 55:1-11

Reviewer #3 (Remarks to the Author):

The manuscript by Shen and coworkers describes a novel iterative approach to construct and validate a quantitative model for a “Weimberg pathway enzyme cascade” through alternate experimental determination of parameters and continuous adaption of a computational model. In a first step, the enzymatic parameters of the respective isolated enzymes were determined and a model was developed that describes the sequential conversion of D-xylose to α ketoglutarate. In a second step, the conversion of D-xylose to α ketoglutarate in a one-pot enzyme cascade with all enzymes was investigated. The parameter determined were used to adapt the previously developed computational model. During these experiments, it became apparent that NADH allosterically inhibits the α ketoglutarate semialdehyde dehydrogenase and that the 2-keto-3-deoxy-xylonate dehydratase is inhibited by D-xylonate, α ketoglutarate semialdehyde and α -ketoglutarate. Based on these observations, the model

was adapted again and applied for optimization of the conversion efficiency within this enzyme cascade. It could be shown that the prediction of optimal enzyme concentrations by the model also improved the conversion efficiency of “older enzymes” (which were first recharacterized with regard to their enzyme activities) in a one pot enzyme cascade.

The manuscript is very well written, easy to follow and the topic in general is very interesting. However, from my point of view several findings are not really new, are in contradiction to results of in vivo experiments, and some conclusions require some further experiments (see below). My point of criticism is that the authors overestimate the usability of their model for establishing the Weimberg pathway in biotechnologically interesting hosts. Reason for this is simply the complexity of the (intra)cellular environment, which cannot be “simulated” in in vitro assays and which changes rapidly within very short time-frames. Nonetheless, some findings such as the need for an efficient NAD⁺ recycling might prove useful.

Major comments:

- Title: The title is inappropriate as it does not reveal anything of the content and reads more like a typical title for a review on this topic.
- Line 35-37: “Thereby, several bottlenecks in pathway performance could be identified, with NADH inhibition of the dehydrogenases, and metal-ion dependence of the dehydratase being most serve.” The authors leave the impression that the metal dependence of the dehydratase is new. However, this has already been shown several times in the past. What is new, however, is the fact that the greatest activity of the tested metal-ions was achieved with manganese ions. Please rephrase.
- Line 85: Here the literature is not up to date. Apparently, scientists made some progress in engineering *C. glutamicum* (Brüsseler, C. et al. (2018) *Bioresour Technol.* 249:953-961; Brüsseler, C. et al. (2019) *Metab Eng Commun*) and very recently establishing the full pathway in *S. cerevisiae* has also been achieved (Borgström, C. et al. (2019). *Metab Eng.* 55:1-11)
- Line 104-106: “For the Weimberg pathway in *C. crescentus*, detailed enzyme kinetic data are only available for the first and the third enzyme,...” Tai and coworkers validated enzymatic activities from three of the five Weimberg pathway enzymes (Tai et al. (2016) *Nat Chem Biol* 12:247–253). These scientists also identified the 2-keto-3-deoxy-xylonate dehydratase (encoded by *xylX*) as the bottleneck enzyme. The authors should discuss the results of this publication. Noteworthy, Tai and coworkers identified a different bottleneck as described in this manuscript!
- Line 134: The kinetic parameters for the enzymes of *Caulobacter crescentus* differ not only absolutely but also relatively from the values determined in a different study (again Tai et al., 2016). This is neither mentioned nor discussed.

- Line 270: It appears a bit odd that the authors focus on the aspect of “stored enzymes”. For convincingly showing that the model is valid, the data from the first experiment with fresh enzyme and optimized abundances should have been included.

Minor comments:

- Line 81: “For whole-cell biocatalytic conversion of D-xylonate the D-xylose dehydrogenase and xylonolactonase were introduced into *Corynebacterium glutamicum*...” Please correct this sentence. By the introduction of a D-xylose dehydrogenase and xylonolactonase nothing would happen to D-xylonate. The authors mean that with the introduction of these enzymes the conversion of D-xylose into D-xylonate was established.

- Line 84: “*Co. glutamicum*” I guess, like always, it should be only the first letter “*C. glutamicum*”.

- Figure 2C: Apparently, the KGSA concentration exceeds the XYL concentration by almost 2 mM, which is not possible.

- Line 425: The lactonase is not needed in *C. glutamicum* and it apparently it is not needed to “speed up” the Weimberg pathway in this bacterium in vivo (Brüsseler, C. et al. (2019)). This should be at least discussed.

Taken together, I think that this manuscript presents some interesting aspects, but is more interesting for readers active in the same field and thus more suitable for a specialized journal.

Reviewer #4 (Remarks to the Author):

Lu Shen et al. provide a comprehensive study about the enzymes involved in the Weimberg pathway in *Caulobacter crescentus* and some approaches for their biotechnological applications. Indeed, the

pathway has high biotechnological potential since it converts xylulose (e.g. from hydrolysates of low-value lignocellulose) into valuable C5 compounds such as α -ketoglutarate. More specifically, the authors purified each of the recombinant enzymes and used a combination of state-of-the-art methods in enzymology to carefully determine the kinetic constants (initial rate constants) and properties (substrate specificities, buffer compositions, pH and temperature optima). ^1H NMR spectroscopy was used to monitor the intermediates/products in a sequential enzyme cascade converting xylose into α -ketoglutarate. This confirmed the knowledge about the pathway and the model calculations performed by the authors. The reaction cascade also worked quite nicely for the first half of the pathway in one-pot reaction mixtures where all five enzymes were added from the beginning showing that the later enzyme KDX dehydratase and KGSA dehydrogenase were inhibited by some components in the mixture, especially KGSA dehydrogenase by NADH. On this basis, the product yields could be improved by the addition of an NAD⁺ recycling system. The experimental data were then used for model construction and validation aimed at highest conversion efficiency. In another round of sequential enzyme cascades (using the optimized protein ratios) the validity of the modeling approach was nicely demonstrated.

The authors also used their methodology with cell-free crude extracts from *C. crescentus*. On this basis, limiting reactions could again be addressed and the conversion rates could be improved by the addition of Mn²⁺ and (again) an NAD⁺ recycling system. Finally, the authors used their knowledge about the XAD enzyme to efficiently synthesize 2-keto-3-deoxy sugar acids at large scale. By combination of the enzyme cascade with a dioxygenase from *B. thuringiensis*, isoleucine could be completely converted into 4-hydroxyisoleucine.

Without doubt, the study is a prime example of modern enzymology providing novel data of the enzymes in the Weimberg pathway and their potential applications in biotechnology. On the other hand, the novelty of these data for microbiology is limited. No doubt that all the methods and experiments were carefully performed and provided correct data. Certainly, in combination of these methods the study could be a kind of gold standard in doing and exploiting enzymology data. However, the individual methods do not provide the highest degree of novelty required for publication in *Nature Communications*. Rather, it appears to be perfectly qualified as a top paper in a more specialized *Nature Journal* or a another top *Journal in Enzymology or Biotechnology* from my point of view.

Some minor issues:

Title: I am not sure whether this rather general title should be used.

NMR analysis: I congratulate the authors for their challenging job to assign and quantify the ^1H NMR signals in even complex mixtures with the isomeric products. The authors mentioned that [^{13}C]xylose

was also used as a substrate that helped to assign the products. I am proposing that in future studies the authors should use [U-13C5]xylose for NMR monitoring.

Response to Referees

The Weimberg Pathway – A combined experimental and modelling approach for pathway optimisation

Lu Shen¹, Martha Kohlhaas², Junichi Enoki³, Roland Meier⁴, Bernhard Schönenberger⁴, Roland Wohlgemuth^{4,5}, Robert Kourist^{3,6}, Felix Niemeyer², David van Niekerk⁷, Christopher Bräsen¹, Jochen Niemeyer^{2*}, Jacky Snoep^{7,8*} and Bettina Siebers^{1*}

First, I would like to thank the reviewers for their overall positive evaluation of our manuscript and also for their critical and constructive comments. We have addressed all of the points in detail, see below, and significantly improved the manuscript and model. We purified all five enzymes in a new isolation and repeated some of the cascade experiments. In addition, we repeated all the cell free extract experiments. We analysed the KGSADH kinetics, specifically with respect to synergistic effects of inhibitors and could remove the factor 30 adaptation that we needed in the previous version.

Finally, we did a thorough rewrite of the manuscript to make the story-line clearer and we feel the document is now even stronger than it was.

Reviewers can login at <https://jij.bio.vu.nl>

Username: shen2019-user-869@jij.bio.vu.nl

Password: qJpsgXwTCW

Username: shen2019-user-665@jij.bio.vu.nl

Password: GRwN2GdL5r

Username: shen2019-user-725@jij.bio.vu.nl

Password: CnwpPGSxpX

SEDMLs (scripts to reproduce the model simulations shown in the manuscript) are available at <https://jij.bio.vu.nl/models/experiments/#review>

Models are available at <https://jij.bio.vu.nl/models/#review>

Reviewers' comments:

Reviewer #1 (Remarks to the Author):

In this paper the authors explore a potentially useful pathway for the breakdown of xylose (a major sugar from cellulose biomass): the Weimberg pathway, a 5 step pathway that converts xylose into a-ketoglutarate. Every step in the pathway is highly favorable thermodynamically at standard state and it is carbon conserving. But so far, the pathway has not worked well in engineered organisms. Here the authors carefully examine the pathway biochemically and find that one enzyme requires Mn^{2+} and other enzymes are inhibited by NADH. By introducing NADH recycling and Mn, they develop a well-functioning pathway in vitro. The lessons learned from a purified system could also be applied to a crude lysate system. There are definitely some very nice aspects of this work, particularly the characterization of the pathway first by basic enzyme kinetic assays, then by following conversions sequentially by NMR which was beautifully done. Yet I have some concerns about the model and what I view as odd choices in presentation.

Specific Comments:

1) The initial one pot reaction without NADH recycling didn't "go to completion." This is attributed to inhibition by various metabolites. Yet the inhibition results seen were quite modest and would not account for the reaction basically stopping. This explanation simply doesn't make sense to me. In fact, the authors had to arbitrarily decrease the activity of KGSADH in their model by a not insignificant 4-fold and increase the inhibition of NADH by a whopping 30-fold to fit the experiment. It seems to me there is something quite wrong with the model. I would like the authors to consider two possibilities: a) A synergistic effect of several metabolites. What happens when you mix them together? Are the effects magnified?

We tested additional inhibitory effects on the KGSADH and observed that in addition to its products (KG and NADH) the enzyme is also inhibited by KDX (see supplementary material, Fig. 8e) and, importantly, that KDX binding affects the NADH inhibition (see supplementary material, Table 7). We included these two inhibitory effects in the rate equation for KGSADH and could use this in the model as is for all the simulations; i.e. we did not need to adapt the inhibition constant from the measured values, so got rid of the factor 30.

b) Or more likely...Is it possible that the inhibition by NADH is not really inhibition of the enzyme, but an equilibrium issue? As pointed out by the authors, under standard state conditions the reactions are highly favorable, but at the lower concentrations that exist in the one pot system it is possible that you are closer to equilibrium. If so, it would make sense that you can't push the reaction further as NADH builds up. An equilibrium issue would also be completely consistent with the fact that NADH recycling makes everything work nicely.

We tested the reversibility of the KGSADH in initial rate experiments, but could not measure activity when incubated with NADH and KG. In addition, the complete conversion of KGSA to KG in the sequential pathway reconstitution (Fig. 2c in the main document), in the presence of high concentrations of NADH suggests that the equilibrium lies far towards the products. Thermodynamic back pressure does not seem to affect the enzyme activity as much as the kinetic binding of products (see answer to 1a).

2) Line 208: "It is unclear why the NADH inhibition is much stronger in the complete pathway analysis, compared to the individual conversion assay and the initial rate kinetics." I don't think you can say that NADH inhibition is stronger in the complete pathway, only that your MODEL

requires it. Other than the fact that it makes the data fit better, there isn't really evidence that that's going on. In fact, your experimental data would suggest otherwise.

Indeed, we should have stated that for the model description to be accurate we need a stronger inhibition of NADH on the KGSADH in the one pot experiment. The issue is not so relevant anymore since we observed that KDX (present in the one-pot experiment but not in the individual conversions) does affect the NADH binding. Thus, in the model we now used the NADH inhibition constant estimated in the initial rate estimation for all the simulations, and the factor 30 was not used anymore.

3) The authors organize the "Model Application for Computational Pathway Design" and the "Model Significance, Usability and Applications" section in a very confused manner in my opinion. They describe how they could computationally optimize the system to generate KG faster. That is not really a finding. You can generate many fantasies in a computer. What the reader wants to know next is does the model actually predict that result correctly, which would be great. But no, we have to go into a discussion of XLA. Then you claim the model has been "validated" starting on line 262, but it hasn't really happened at that point in the story. You have to wait until the next section to find out that in fact the model seems to do OKish. Throughout these sections it is very hard for the reader to know if you are talking about computational results or experimental results. I think this entire presentation needs to be rearranged.

We apologize for our apparent unclear presentation of the model application sections, the structure reflected the chronological way in which the project was done, first making model predictions and subsequent experimental tests. Although this nicely illustrates the importance of the model for our experimental design, this might not be the best way to communicate our findings. We have adapted the sections.

4) The section on the Significance of XLA seems rather inane to me. The reaction can occur spontaneously which is nice to know, but the fact that when the other enzymes are fast enough the spontaneous reaction is rate limiting is simply obvious and doesn't require a computational model. I think this section should be deleted and replaced with a one sentence summary, if that.

Here we do not agree with the reviewer, the question whether the spontaneous reaction rate is fast enough for the metabolic pathway to function properly is an open question (as also indicated by reviewer 2). The enzyme is essential for growth of *C. crescentus* on D-xylose, but in metabolic engineering approaches the enzyme is sometimes not included, which can be beneficial as it decreases the toxic D-xylonate accumulation resulting from unbalanced reaction rates, but would become limiting at higher pathway fluxes.

Based on the other reviewers' comments on the experiment, and also because we think it is important for model validation to test a wide variety of conditions, we decided to keep the XLA deletion incubation in the main document. Leaving out an enzyme in the pathway is a good model validation test as it has a drastic effect on the accumulation of its substrate, which was accurately predicted by the model. And yes, one does not need a model to predict that the substrate of XLA would increase upon its omission in the incubation, but without a model it would be impossible to predict the extent and dynamics of the accumulation, and whether it would have an effect on the KG production rate.

5) I'm impressed that the predicted optimization by the model was reflected pretty well in the experiment of Fig. 3D. THIS is a nice validation of the model! Yet the authors manage to

confuse this point by talking about how they used old enzymes that were somewhat inactivated over time. This is not interesting. Enzymes die over time and some enzyme preps are more active than others so you need to adjust for their specific activity. We all know that. I don't understand why you would swallow the really nice result with what seems to me an inane tangent.

We were also glad about the strong improvement of KG production, and the predictive strength of the model for the optimised conditions. Based on the suggestions by the other reviewers we repeated the optimised incubation with newly isolated enzymes, and the results of the different incubations are in excellent agreement. For us the "old" enzyme incubation was important to show the versatility of the model, simply enter the changed specific activities and the model will predict the optimised conditions. The detailed information is now transferred to the supplementary information.

6) The results in Fig. 3D are very strange. At 100 min, there is apparently complete conversion to KG, yet there's still 1 mM KDX present, which slowly disappears. Something doesn't add up. And where is it going?

We thank the reviewer for the comment. We agree that KDX should not be present when full conversion has already been reached. The KDX-quantification is very sensitive to the baseline of the NMR-spectra and there are three different KDX species that have to be considered. In the reported experiment, we did not observe all three species and thus have used a correction factor to account for the others. Together with a non-ideal baseline, this suggested the presence of KDX, although the observed signals were indeed very small. Thus, we believe that this was indeed an integration problem in the NMR.

To double-check this, we repeated this experiment in duplicate (see figure 3) and indeed, we could no longer observe the presence of KDX at all. Thus, in the optimized cascade, there is no KDX building up.

7) The biotechnology section is really strange.

a) In one application the authors use a single enzyme from the Weimberg pathway to convert a handful of sugars. Using a single enzyme and producing gram quantities of material is as old as biochemistry. Why should anyone be impressed by that?

Our aim with the biotechnological applications was to show the importance of the Weimberg pathway enzymes for commercial production of intermediates. In itself the use of enzymes in biotechnology is of course not new, but the versatility of the XAD to produce gram levels of stereospecific pure products at >95% purity, starting from cheap substrates has enormous commercial value. So for one, Sigma-Aldrich/Merck was very impressed and is using the application for the commercial production of KDG, KDX, and KDGal. Since all the reactions in the Weimberg pathway run to completion the same approach can be used to produce any of the other intermediates. Availability of the intermediates is important for other production processes, but also for scientific research on the pathway enzymes.

b) In the second application they say that they use the Weimberg pathway to generate KG, which can then be used as a sacrificial substrate in the production of hydroxyisoleucine. This seems to be more of an interesting application, but it was completely unclear what they did. They show a figure in the supplement with results, but there is no experimental description anywhere in the paper or the supplement. A supplementary figure showing the system set up and experimental details are essential if this is to be included in the paper.

For the second application we used the Weimberg pathway cascade to first produce KG, and subsequently converted this to hydroxyisoleucine. We apologize for the confusion the information was at the end of the Method part and we now transferred it to the biotechnology part in the supplementary information.

8) I can't figure out what the different colored lines are in Suppl. Fig. 8d and what the vertical axis means.

In Figure 8c and 8d we show the inhibitory effect of KG and NADH on the KGSADH, for this we used different substrate concentrations, 1 or 2 mM KGSA in Fig 8c, and 5, 1, 0.5, and 0.05 mM NAD in Fig 8d. The different substrate concentrations are indicated with different colours. For Fig. 8d we forgot to indicate the concentrations of the substrate in the legend, which we have corrected now. In addition, we have included the KDX inhibition on the enzyme in an additional panel Fig. 8e.

For easier comparison we have shown the enzyme activities as a fraction of the maximal rate of the enzyme (v/V_m), this can be seen as a percentage activity, i.e. a fraction of 1 indicates an activity equal to 100% of the V_m .

9) I have no idea what Suppl. Fig. 12 is supposed to convey other than the fact that if you vary parameters in the model you get different results. There is no experimental data shown for comparison.

With Fig. 12 we wanted to show the effect of changing the two model parameters for which we had no direct experimental confirmation, i.e. the factor 4 for the KDXD and the factor 30 for the binding constant of NADH to the KGSADH. Since we could resolve the factor 30, via inhibition studies in initial rate experiments, the figure is not so relevant anymore and we have decided to leave it out.

Minor things:

10) The title should say more about what the paper is about or what the authors want to communicate. Maybe something like, "Using cell free modeling to identify and correct bottlenecks in the Weimberg Pathway for conversion of cellulosic biomass."? We changed the title according to the suggestions of the reviewer to "The Weimberg Pathway – A combined experimental and modelling approach for pathway optimisation"

11) I believe KGSADH can use BOTH NAD⁺ and NADP⁺? It's not clear why you include that in Fig. 1 since it is rather confusing when you discuss NADH inhibition for a reaction that seems to use NADP⁺ according to the figure. I understand that you want to be complete, but maybe you could state the fact that it's non-specific in the figure legend, but leave it as NAD⁺ in the actually figure? Or maybe it's just me.

We adjusted Fig. 1 according to the suggestions just showing NAD⁺.

Reviewer #2 (Remarks to the Author):

In the manuscript of Shen and coworkers the Weimberg pathway from *C. crescentus* was studied in detail by applying classical in vitro enzyme assays, sophisticated NMR analysis for intermediate dynamics and standard mechanistic pathway modeling. Novel results concern the inhibition patterns of selected enzymes by various pathway intermediates as well as the production of two new compounds using a single biocatalytic step or the extended Weimberg pathway, respectively. The topic is very interesting and the paper is well written. However, some of the claims are either not new or need further experimental support.

Major remarks:

1) L85: Recently, several studies showed improved performance of the Weimberg pathway in prokaryotic and eukaryotic organisms (Ref. 1-4). These should be taken into account for discussion.

We apologize for not being complete in our literature references, we have scanned the literature again, and in addition have included the references indicated by the reviewer, and integrated the findings reported in the manuscripts in our discussion.

2) When applying the one-pot cascade, the model fails to accurately simulate the measured trajectories for XLAC, XA (both Fig. 3A-D) and KDX (Fig. 3D). Please provide measures for goodness of fit (e.g. RMSE) for all model fits.

We agree that it is important to include an error estimate for the experimental data and for the model simulations. For the revised manuscript, all enzymes were newly isolated and all one-pot cascade experiments were repeated and the standard error of the mean for the experimental data is shown in all figures. In addition, we show an estimate for the model accuracy by performing Monte Carlo simulations where a 10% error on model parameters and 5% error in initial substrate concentrations were analysed (see supplementary information for detail). The error estimates for the model simulations are indicated with colour-shaded bands around the mean value.

We would like to point out that the simulations in Fig. 3 B-D are not model fits, but model predictions (a single parameter was adapted for Fig 3A, but no other model adaptations were made). As far as model predictions go, and note that all parameters except for 1 were estimated on direct experimental data, not fitted to systemic behaviour, the predictions are very close for a wide set of perturbations.

3) What was the rationale for applying only simple Convenience kinetics to model each enzymatic step? A comparison of different kinetic approaches including Michaelis-Menten type and approximative formats would have substantially improved the modeling part.

We do not quite understand the comment of the reviewer. All the rate equations are based on Michaelis Menten kinetics, and include product inhibition based on competitive binding with the substrate.

As can be seen in the fitting of the rate equations to the initial rate kinetics (Figs 4-8 in the supplementary material), the equations can describe the experimental data very well, and there is no need for more elaborate kinetic mechanisms.

4) L234ff: For convincingly showing that the model is valid, the data from the first experiment with fresh enzyme and optimized abundances should have been included as Fig. 3D. The data from the 2nd experiment using long-term stored enzymes might be included in the

Supplement. However, apart from KG, the model fits seem to be not very accurate and therefore the predictive power of the model is questionable.

We purified the enzymes again, and repeated the one-pot cascades, the figures show averages for the different repeats, including experimental error. In addition, we added KDX inhibition of the KGSADH to the model. The model predictions have improved significantly. We would like to stress again that these simulations are model predictions, not model fits. Considering that the model was not adapted for the simulations shown in (Fig 3. b-d, and Fig. 4 b-d), the predictive power of the model for the different perturbations, including the cell free extract incubations is very good (see also reviewer 1 (5)).

5) L309: No metabolite quenching was applied to immediately stop enzymatic conversion. Please justify.

Upon sampling the proteins were directly precipitated in trichloroacetic acid, which stops the enzymatic reaction immediately (see material part for details).

6) L320: Again, the model fits are not really accurate and proper statistics are missing!

In the revised manuscript, we have included experimental error bars, and bands that indicate confidence regions for model simulations assuming a 10% error in parameter estimations. The model was extended to include KDX inhibition of KGSADH (Fig. 8e in revised version), and we analysed the cell free extract for XAD activity (at the respective protein concentration in the D-xylose conversion assay 0.4 mg/ml) and its dependence on Mn^{2+} more carefully (see Supplementary materials 2.2.3, Table 10 and 11, fig).

Now, the model predictions for the cell free extract simulations are very good.

7) L424, L430: Recently, Brüsseler et al. could already prove experimentally that a lactonase is not required and, most importantly, not limiting for *in vivo* Weimberg pathway operation.

We thank the reviewer for pointing us to this new publication. The relevance of the lactonase for *in vivo* Weimberg pathway operation is interesting and not quite resolved. In the Brüsseler paper the reviewer pointed out (Brüsseler et al., 2019) it is shown that recombinant expression of the enzyme is not important for the pathway flux in *Corynebacterium glutamicum*, but the authors did not check whether endogenous lactonase activity is present in the extract (note that they did discover a new native KGSADH in the strain, Cg0535). On the other hand, in an earlier study (Stephens et al., 2007) it was shown that for *Caulobacter crescentus* all five enzymes in the pathway are essential for growth on xylose.

Interestingly our cascade analysis shows that the pathway could operate fine without the enzyme under the reference conditions, i.e. not limiting the KG production rate, but would become limiting under optimised conditions. This indicates that the role of the lactonase is condition dependent, leading to the observed differences for importance of the enzyme *in vivo*. We discuss this in more detail in the revised manuscript.

Minor remarks:

8) The title is rather uninformative and should be more specific, e.g. "In vitro studies on enzymes from *Caulobacter crescentus* Weimberg pathway"

We changed the title to "The Weimberg Pathway – A combined experimental and modelling approach for pathway optimisation"

9) Fig. 2C: The model fits for the XAD step are rather bad. The linear increase of KDX points to enzyme inhibition and the model already covers KDX and NADH as effectors. Please discuss.

We agree that the model prediction for the XAD reaction is not as good as the prediction for the other steps. We are unsure what causes this, the constant activity leading to a linear change in substrate and product concentrations indicates that neither the substrate decrease, nor product increase are responsible for inhibiting the reaction during the conversion, but points at a generally lower activity. We noticed that the enzyme is quite sensitive to the assay conditions, i.e. with respect to metal ions, and also dependent on total protein concentration, but we are unsure what caused the enzyme activity to be slower in the conversion assay compared to the initial rate kinetics.

10) Fig. 2C: In one experiment there was an accumulation of KGSA over 6 mM, which is significantly higher than the initial 5 mM xylose. Please discuss.

The integration of the relatively low signals in the NMR spectra leads to significant experimental error, which is now made apparent by including the experimental error bars. We repeated the one-pot cascade experiments again and the mean value for the different experiments is now closer to the maximal attainable value, but the relatively large error bars remain.

References:

- 1) Radek, A. et al. (2017) Miniaturized and automated adaptive laboratory evolution: Evolving *Corynebacterium glutamicum* towards an improved D-xylose utilization. *Bioresour Technol* 245(Pt B):1377-1385
- 2) Brüsseler, C. et al. (2018) The myo-inositol/proton symporter IolT1 contributes to D-xylose uptake in *Corynebacterium glutamicum*. *Bioresour Technol.* 249:953-961
- 3) Brüsseler, C. et al. (2019) Alone at last! - Heterologous expression of a single gene is sufficient for establishing the five-step Weimberg pathway in *Corynebacterium glutamicum*. *Metab Eng Commun.* doi: 10.1016/j.mec.2019.e00090
- 4) Borgström, C. et al. (2019) Identification of modifications procuring growth on xylose in recombinant *Saccharomyces cerevisiae* strains carrying the Weimberg pathway. *Metab Eng.* 55:1-11

Reviewer #3 (Remarks to the Author):

The manuscript by Shen and coworkers describes a novel iterative approach to construct and validate a quantitative model for a “Weimberg pathway enzyme cascade” through alternate experimental determination of parameters and continuous adaption of a computational model. In a first step, the enzymatic parameters of the respective isolated enzymes were determined and a model was developed that describes the sequential conversion of D-xylose to α ketoglutarate. In a second step, the conversion of D-xylose to α ketoglutarate in a one-pot enzyme cascade with all enzymes was investigated. The parameter determined were used to adapt the previously developed computational model. During these experiments, it became apparent that NADH allosterically inhibits the α ketoglutarate semialdehyde dehydrogenase and that the 2-keto-3-deoxy-xylonate dehydratase is inhibited by D-xylonate, α ketoglutarate semialdehyde and α -ketoglutarate. Based on these observations, the model was adapted again and applied for optimization of the conversion efficiency within this enzyme cascade. It could be shown that the prediction of optimal enzyme concentrations by the model also improved the conversion efficiency of “older enzymes” (which were first recharacterized with regard to their enzyme activities) in a one pot enzyme cascade.

The manuscript is very well written, easy to follow and the topic in general is very interesting. However, from my point of view several findings are not really new, are in contradiction to results of in vivo experiments, and some conclusions require some further experiments (see below). My point of criticism is that the authors overestimate the usability of their model for establishing the Weimberg pathway in biotechnologically interesting hosts. Reason for this is simply the complexity of the (intra)cellular environment, which cannot be “simulated” in in vitro assays and which changes rapidly within very short time-frames. Nonetheless, some findings such as the need for an efficient NAD⁺ recycling might prove useful.

Major comments:

1) Title: The title is inappropriate as it does not reveal anything of the content and reads more like a typical title for a review on this topic.

We changed the title to “The Weimberg Pathway – A combined experimental and modelling approach for pathway optimisation”

2) Line 35-37: “Thereby, several bottlenecks in pathway performance could be identified, with NADH inhibition of the dehydrogenases, and metal-ion dependence of the dehydratase being most serve.” The authors leave the impression that the metal dependence of the dehydratase is new. However, this has already been shown several times in the past. What is new, however, is the fact that the greatest activity of the tested metal-ions was achieved with manganese ions. Please rephrase.

The sentence is changed to: “Particularly the first dehydratase, which known metal-ion dependency would quickly make the enzyme rate limiting when not supplemented with Mg²⁺ or Mn²⁺, leading to accumulation of xylonate.”

3) Line 85: Here the literature is not up to date. Apparently, scientists made some progress in engineering *C. glutamicum* (Brüsseler, C. et al. (2018) *Bioresour Technol.* 249:953-961; Brüsseler, C. et al. (2019) *Metab Eng Commun*) and very recently establishing the full pathway in *S. cerevisiae* has also been achieved (Borgström, C. et al. (2019). *Metab Eng.* 55:1-11)

We apologize for not having included the newest references on the Weimberg pathway. We have now included these references, and discuss our results taking these new studies into account.

4) Line 104-106: “For the Weimberg pathway in *C. crescentus*, detailed enzyme kinetic data are only available for the first and the third enzyme,...” Tai and coworkers validated enzymatic activities from three of the five Weimberg pathway enzymes (Tai et al. (2016) *Nat Chem Biol* 12:247–253). These scientists also identified the 2-keto-3-deoxy-xylonate dehydratase (encoded by *xylX*) as the bottleneck enzyme. The authors should discuss the results of this publication. Noteworthy, Tai and coworkers identified a different bottleneck as described in this manuscript!

Again we are sorry that we missed this important publication. We added the information in the paper and state that also the KDXD has been studied previously. We compare our results regarding the enzyme kinetic parameters with theirs and also elaborate on the bottleneck including metabolic control analysis (MCA) in the revised version of our manuscript, see also the answer to the next query.

We carefully studied the enzyme assays performed by Tai et al.. Most of the activities (kcat-values) are significantly lower than in our studies. However, there are several differences compared to our study so that it is difficult to compare our data. For example all enzymes were frozen (according to our study some enzymes are more stable at 4°C; we used freshly prepared enzyme for characterization), they used a TRIS buffer pH 7.5 (in our hands HEPES buffer pH 7.5 was more suitable), they used different assays (e.g. the semicarbazid assay, we used the TBA assay) and finally they performed all assays at 30°C and we at 37°C.

5) Line 134: The kinetic parameters for the enzymes of *Caulobacter crescentus* differ not only absolutely but also relatively from the values determined in a different study (again Tai et al., 2016). This is neither mentioned nor discussed.

As discussed above the assays for the three enzymes XDH, XAD and KDXD were performed under different assay conditions and the determined Kcat as well as Km values were therefore significantly different (see above for details).

6) Line 270: It appears a bit odd that the authors focus on the aspect of “stored enzymes”. For convincingly showing that the model is valid, the data from the first experiment with fresh enzyme and optimized abundances should have been included. **We performed a new enzyme isolation and repeated the incubation under the optimised conditions in duplicate. The results obtained with the new isolates were essentially the same as obtained before, and all the experiments are combined in the new Fig. 3D, with error bars indicating the standard error of the mean (see also reviewer 1 (5) and reviewer 2 (4)).**

Minor comments:

7) Line 81: “For whole-cell biocatalytic conversion of D-xylonate the D-xylose dehydrogenase and xylonolactonase were introduced into *Corynebacterium glutamicum*...” Please correct this sentence. By the introduction of a D-xylose dehydrogenase and xylonolactonase nothing would happen to D-xylonate. The authors mean that with the introduction of these enzymes the conversion of D-xylose into D-xylonate was established.

We corrected the error and changed the sentence to: “For whole-cell biocatalytic conversion of D-xylose to D-xylonate the D-xylose dehydrogenase and xylonolactonase were introduced.”

8) Line 84: “Co. glutamicum” I guess, like always, it should be only the first letter “C. glutamicum”.

It is sometimes preferred to use different abbreviations if two organisms that are discussed in the manuscript have the same first letter, i.e. *Caulobacter* and *Corynebacterium*, but we are happy to change this.

9) Figure 2C: Apparently, the KGSA concentration exceeds the XYL concentration by almost 2 mM, which is not possible.

This was also observed by reviewer 2 (6), we have repeated the experiment, re-analysed the NMR data, and indicated the experimental error estimate to the data points.

10) Line 425: The lactonase is not needed in *C. glutamicum* and it apparently it is not needed to “speed up” the Weimberg pathway in this bacterium in vivo (Brüsseler, C. et al. (2019)). This should be at least discussed.

We have now discussed these findings for *C. glutamicum* and contrast them to the findings obtained for *Caulobacter crescentus* (Stephens et al 2007, Genetic Analysis of a Novel Pathway for D-Xylose Metabolism), where all five genes including the lactonase are essential for growth on D-xylose (see also above in our answers to queries by reviewer 1 (4) and 2 (7)). Since no enzyme assays are performed in *C. glutamicum* we cannot exclude that an endogenous lactonase is present. We demonstrate in our study that the role of the lactonase is dependent on the flux through the pathway; it becomes limiting in the optimized state of the enzyme cascade but has no effect on the KG production in our reference condition.

11) Taken together, I think that this manuscript presents some interesting aspects, but is more interesting for readers active in the same field and thus more suitable for a specialized journal.

In addition to reporting on the important findings for the Weimberg pathway, we have put more emphasis on the generic modelling approach for the construction, validation, and application of the mathematical model. The multiple iteration steps between experiment and model to estimate and validate the model parameters, together with the transparent and completely reproducible workflow, for which all data sets and modelling scripts are made available, set a standard for these types of studies.

Reviewer #4 (Remarks to the Author):

Lu Shen et al. provide a comprehensive study about the enzymes involved in the Weimberg pathway in *Caulobacter crescentus* and some approaches for their biotechnological applications. Indeed, the pathway has high biotechnological potential since it converts xylulose (e.g. from hydrolysates of low-value lignocellulose) into valuable C5 compounds such as α -ketoglutarate. More specifically, the authors purified each of the recombinant enzymes and used a combination of state-of-the-art methods in enzymology to carefully determine the kinetic constants (initial rate constants) and properties (substrate specificities, buffer compositions, pH and temperature optima). ^1H NMR spectroscopy was used to monitor the intermediates/products in a sequential enzyme cascade converting xylose into α -ketoglutarate. This confirmed the knowledge about the pathway and the model calculations performed by the authors. The reaction cascade also worked quite nicely for the first half of the pathway in one-pot reaction mixtures where all five enzymes were added from the beginning showing that the later enzyme KDX dehydratase and KGSA dehydrogenase were inhibited by some components in the mixture, especially KGSA dehydrogenase by NADH. On this basis, the product yields could be improved by the addition of an NAD⁺ recycling system. The experimental data were then used for model construction and validation aimed at highest conversion efficiency. In another round of sequential enzyme cascades (using the optimized protein ratios) the validity of the modeling approach was nicely demonstrated.

The authors also used their methodology with cell-free crude extracts from *C. crescentus*. On this basis, limiting reactions could again be addressed and the conversion rates could be improved by the addition of Mn²⁺ and (again) an NAD⁺ recycling system. Finally, the authors used their knowledge about the XAD enzyme to efficiently synthesize 2-keto-3-deoxy sugar acids at large scale. By combination of the enzyme cascade with a dioxygenase from *B. thuringiensis*, isoleucine could be completely converted into 4-hydroxyisoleucine.

Without doubt, the study is a prime example of modern enzymology providing novel data of the enzymes in the Weimberg pathway and their potential applications in biotechnology. On the other hand, the novelty of these data for microbiology is limited. No doubt that all the methods and experiments were carefully performed and provided correct data. Certainly, in combination of these methods the study could be a kind of gold standard in doing and exploiting enzymology data. However, the individual methods do not provide the highest degree of novelty required for publication in *Nature Communications*. Rather, it appears to be perfectly qualified as a top paper in a more specialized *Nature Journal* or another top *Journal in Enzymology or Biotechnology* from my point of view.

We thank the reviewer for this positive evaluation of our paper. Although the reviewer is correct that the methods we have used are in their own right not new, the strength of the manuscript lies in the unique combination of experimental methods and iteration with the mathematical model. The set of experiments allow for independent estimation of first substrate saturation and V_{max} values, then the product sensitivity and finally allosteric regulation. In addition, we have strictly separated model construction from model validation and application. Finally, all data files and models are available for the readers and the figures can be life reproduced ensuring transparency and reproducibility of the modelling process.

This method (summarised in Fig. 5) is generic and could indeed be seen as an excellent standard method for model construction and validation. We have stressed this generic modelling aspect of the manuscript somewhat more in the revised version.

Some minor issues:

1) Title: I am not sure whether this rather general title should be used.

We changed the title to "The Weimberg Pathway – A combined experimental and modelling approach for pathway optimisation"

2) NMR analysis: I congratulate the authors for their challenging job to assign and quantify the ^1H NMR signals in even complex mixtures with the isomeric products. The authors mentioned that $[1-^{13}\text{C}]$ xylose was also used as a substrate that helped to assign the products. I am proposing that in future studies the authors should use $[\text{U}-^{13}\text{C}_5]$ xylose for NMR monitoring.

We thank the reviewer for this nice suggestion. Indeed, we have tried to use $[\text{U}-^{13}\text{C}_5]$ -xylose, but found that this makes the ^{13}C analysis more difficult, since this leads to coupling between the ^{13}C centers. Thus, we observed complex multiplets for each carbon center instead of 5 singlets. For this reason, we resorted to the use of $[1-^{13}\text{C}]$ -xylose, which give one defined singlet for C-1 in the ^{13}C NMR.

Reviewer #1 (Remarks to the Author):

The authors have addressed my concerns and it reads much more clearly now. I still think the biotechnology section feels tacked on since it is hard to see how the results from the rest of the paper impact this section, but we can agree to disagree.

Reviewer #2 (Remarks to the Author):

All my concerns have been completely dispelled and the paper might be published as it is.

Reviewer #3 (Remarks to the Author):

All my points were addressed, nonetheless I still think that the "sum of novelties" is quite low. However, if the other reviewers come to the conclusion that this manuscript should be published in this journal, I will support this.

Reviewer #4 (Remarks to the Author):

All of my concerns have been addressed in the authors' revision. The manuscript can now be accepted for publication in Nature Communications.

Response to Referees

The Weimberg Pathway – A combined experimental and modelling approach for pathway optimisation

Lu Shen¹, Martha Kohlhaas², Junichi Enoki³, Roland Meier⁴, Bernhard Schönenberger⁴, Roland Wohlgemuth^{4,5}, Robert Kourist^{3,6}, Felix Niemeyer², David van Niekerk⁷, Christopher Bräsen¹, Jochen Niemeyer^{2*}, Jacky Snoep^{7,8*} and Bettina Siebers^{1*}

We like to thank the reviewers for their overall positive evaluation of the revised manuscript and the (in principle) acceptance of our manuscript. We appreciate the time and effort of the reviewers; their critical and constructive comments were very helpful for us in order to further improve the manuscript.

REVIEWERS' COMMENTS:

Reviewer #1 (Remarks to the Author):

The authors have addressed my concerns and it reads much more clearly now. I still think the biotechnology section feels tacked on since it is hard to see how the results from the rest of the paper impact this section, but we can agree to disagree.

Reviewer #2 (Remarks to the Author):

All my concerns have been completely dispelled and the paper might be published as it is.

Reviewer #3 (Remarks to the Author):

All my points were addressed, nonetheless I still think that the "sum of novelties" is quite low. However, if the other reviewers come to the conclusion that this manuscript should be published in this journal, I will support this.

Reviewer #4 (Remarks to the Author):

All of my concerns have been addressed in the authors' revision. The manuscript can now be accepted for publication in Nature Communications.